# Therapeutic senescence via GPCR activation in synovial fibroblasts facilitates resolution of arthritis

Trinidad Montero-Melendez [1,2]*, Ai Nagano [3], Claude Chelala[3,4], Andrew Filer [5], Christopher D. Buckley [5,6] & Mauro Perretti [1,2]*

Rheumatoid arthritis affects individuals commonly during the most productive years of adulthood. Poor response rates and high costs associated with treatment mandate the search for new therapies. Here we show that targeting a specific G-protein coupled receptor promotes senescence in synovial fibroblasts, enabling amelioration of joint inflammation. Following activation of the melanocortin type 1 receptor ($MC_1$), synovial fibroblasts acquire a senescence phenotype characterized by arrested proliferation, metabolic re-programming and marked gene alteration resembling the remodeling phase of wound healing, with increased matrix metalloproteinase expression and reduced collagen production. This biological response is attained by selective agonism of $MC_1$, not shared by non-selective ligands, and dependent on downstream ERK1/2 phosphorylation. In vivo, activation of $MC_1$ leads to anti-arthritic effects associated with induction of senescence in the synovial tissue and cartilage protection. Altogether, selective activation of $MC_1$ is a viable strategy to induce cellular senescence, affording a distinct way to control joint inflammation and arthritis.

[1] The William Harvey Research Institute, Barts and The London School of Medicine, Queen Mary University of London, Charterhouse Square, London EC1M 6BQ, UK. [2] Centre for Inflammation and Therapeutic Innovation, Queen Mary University of London, London, UK. [3] Barts Cancer Institute, Barts and The London School of Medicine, Queen Mary University of London, Charterhouse Square, London EC1M 6BQ, UK. [4] Life Sciences Initiative, Queen Mary University of London, London, UK. [5] NIHR Birmingham Biomedical Research Centre, University Hospitals Birmingham NHS Foundation Trust and University of Birmingham, Institute of Inflammation and Ageing, Birmingham, UK. [6] Kennedy Institute of Rheumatology, Nuffield Department of Orthopaedics, Rheumatology and Musculoskeletal Sciences, University of Oxford, Oxford, UK. *email: t.monteromelendez@qmul.ac.uk; m.perretti@qmul.ac.uk

From Latin *senēscere*—to grow old—the process of cellular senescence is a state of irreversible growth arrest which serves as a potent tumor suppressor mechanism. Although their ultimate fate is death, senescent cells remain highly metabolically active for long periods and acquire a characteristic phenotype, referred as the senescence-associated secretory phenotype (SASP) leading to context dependent outcomes[1]. Senescence is a strategy favored by natural selection, evidencing its positive protective value. However, senescence can also cause tissue dysfunction typical with ageing, encouraging the development of therapeutics based on senolysis or SASP reprogramming as life-extension approaches[2]. The existence of these two faces of senescence, one life-saving and another detrimental, is usually explained by the antagonistic pleiotropy theory, suggesting that ageing sits in the shadow of selection where harmful late-acting processes can remain in a population as long as they confer reproductive fitness early in life, as later individual performance is irrelevant for natural selection forces.

More recently, in addition to cancer and ageing, a prominent role for senescence in tissue repair has emerged, particularly when affecting fibroblasts. Expression profiling analysis revealed that senescent fibroblasts get locked in an activated state that mimics the early remodeling phase of wound healing[3]. Later, it has been reported that senescent fibroblasts accelerate would closure via PDGF-AA secretion[4] and facilitate the reversion of fibrosis in a mouse model of liver cirrhosis[5]. In the human synovium, fibroblasts are key effectors of tissue destruction in rheumatoid arthritis (RA). Fibroblast main role is to provide structural support, lubrication to allow low friction movements of the articular joints and nutrients to the avascular cartilage. This benign scenario changes dramatically during RA[6,7]. The synovial layer transforms into a hyperplastic invasive tissue, expanding up to 10–20 cells thick in which synovial fibroblasts (SFs) acquire an aggressive proliferative phenotype turning into cytokines factories, driving cartilage and bone destruction, and promoting sustained recruitment and retention of immune cells. Thus, SFs do not merely respond to inflammatory stimuli, but become active aggressors in the RA joints.

The aggressive behavior of SFs in RA can be a consequence of failure of resolution of inflammation. While the resolution of inflammation may be perceived as a separate entity from the pro-inflammatory phase, pro-resolving mechanisms and mediators are activated in parallel to pro-inflammatory mechanisms[8,9]. SFs from RA patients retain their aggressive phenotype in the absence of any inflammatory stimulus[7]. Although SF activation lies downstream of the hierarchy of events that trigger RA, these involving predominantly immune cell infiltration, their failure to switch off after stimulation contributes to the destructive chronic inflammation typical of the RA joint[6]. Targeting SF may represent a promising strategy for reverting chronicity and inducing long-term remission, without inducing immunosuppression.

Melanocortin (MC) receptor agonists are promising candidates for the treatment of inflammatory conditions, including colitis[10], myocardial and cerebral ischemia[11,12], Alzheimer[13], multiple sclerosis[14], and RA[15]. The MC agonist adrenocorticotropin hormone (ACTH) is effective in human RA as shown over 60 years ago[16]. There is more recent evidence that MC therapy leads to protective actions including reduced fibroblast activation, osteoblast proliferation, and improved chondrocyte function[17–19]. In vivo, the synthetic peptide D-Trp[8]-γMSH reduces clinical signs of disease in experimental inflammatory arthritis[20] and urate crystal peritonitis[21]. In addition, the pan-MC agonist AP214 and the biased AP1189 small molecule also display anti-arthritic properties[15,22]. In the present study, we aimed to investigate the potential therapeutic properties of MC drugs on SF from RA patients. Interestingly, we observe that selective activation of the melanocortin receptor $MC_1$ on SF promotes cellular senescence. This receptor, with a non-redundant role in skin pigmentation, is endowed with anti-inflammatory actions[23]. $MC_1$-mediated senescence leads to remarkable therapeutic consequences: cessation of SF proliferation and acquisition of a pro-repair phenotype that highly resembles that observed during senescence-driven wound healing[3–5]. This effect is achieved by activation of the receptor by a small-molecule agonist, and it is the first reported evidence of cellular senescence induced through direct activation of a G-protein coupled receptor (GPCR).

## Results

**The MC system is functional in SFs**. We examined expression and functionality of the MC system in human primary joint SF (Supplementary Table 1). Receptors (*MC1R, MC2R, MC3R, MC4R, MC5R*), ligands, and precursors (*POMC, ASIP, AGRP*), processing enzymes (*PCSK1, PCSK2*), and accessory proteins (*MRAP1, MRAP2*) were studied. *MC1R, MC3R, MC4R*, and *MC5R* receptors were detected at the mRNA level, together with other members of the MC pathway (Fig. 1a). *MC1R* displayed the highest expression among the four receptors (Fig. 1b). SF could release the endogenous anti-inflammatory peptide ACTH (Fig. 1c). With respect to receptor activation, the natural pan-agonist αMSH and the $MC_1$-selective compound BMS-470539 (BMS)[24] induced ERK-phosphorylation (Fig. 1d), although they did not elevate cAMP, the canonical pathway described for this family of GPCRs[25] (Fig. 1e). Both αMSH and BMS raised intracellular $Ca^{2+}$ (Fig. 1f). Functionally, MC receptor activation did not result in remarkable differences on early cellular responses (24–48 h) including scratch gap closure, cytokine release, SF migration or invasion (Fig. 1g–i), though some of the modest effects reached statistical significance.

**Selective $MC_1$ activation induces senescence via ERK1/2**. Both αMSH and BMS reduced 7-day cell proliferation (Fig. 2a). However, cells treated with BMS exhibited a number of specific features including expansion of the lysosomal compartment, presence of bi-nucleated cells, increase in cellular activity—Alamar blue assay—and increases in the extent of β-galactosidase-positive staining (Supplementary Fig. 1A–D). All these outcomes suggested induction of cellular senescence, a feature that was confirmed in a concentration–response assay (Fig. 2b). Interestingly, the pan-agonist αMSH, even when tested for prolonged periods up to 2 weeks, did not produce any of these changes; similar inefficacy was observed with the dual $MC_1/MC_3$ agonist, [D-Trp[8]]-γMSH (Supplementary Fig. 1D). The tumor suppressor protein p53 was up-regulated by BMS treatment (Fig. 2c), together with the senescence marker p16[INK4], whose expression correlated with senescence-associated β-galactosidase (SA-βGal) (Fig. 2d). $MC_1$ selective activation also induced p16[INK4] expression in SF grown as 3D organoids (Fig. 2e).

To establish specificity, experiments were conducted in other cell types. Indeed, these pro-senescence effects downstream selective $MC_1$ activation were not replicated with human primary macrophages (Fig. 2f) or murine melanocytes (Fig. 2g). On the latter cell type, BMS was a potent inducer of melanin synthesis (Supplementary Fig. 1E, F); as predicted this effect was not observed in SF, denoted by lack of melanin production and tyrosinase gene expression (Supplementary Fig. 1G, H). Interestingly, BMS also induced senescence in human dermal fibroblasts (Supplementary Fig. 1I), indicating that (i) this distinct mechanism may be of broader biological significance and (ii) this observation may be of translational value for skin conditions characterized by aberrant fibroblast proliferation and activation.

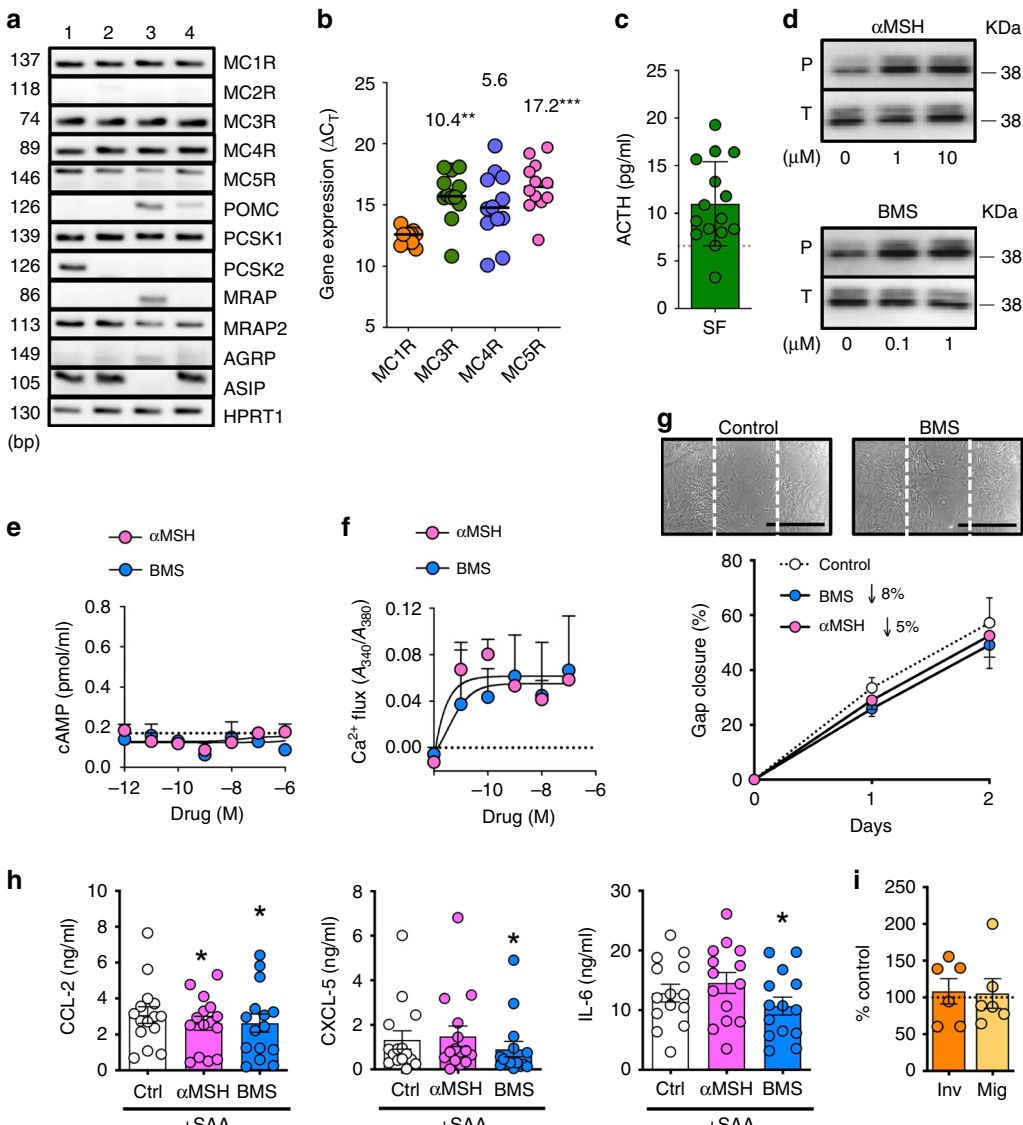

**Fig. 1 The Melanocortin (MC) system in synovial fibroblasts. a** Expression of several components of the MC pathway in RA SF fibroblasts as determined by end-point PCR using 1 μg of RNA ($n = 4$). **b** MCRs expression was further analyzed by real-time PCR using 1 μg of RNA. Cycle threshold ($C_T$) values normalized against the reference control *HPRT1* are shown (lower $C_T$ values denote higher expression). Numbers above bars indicate overexpression of *MC1R* compared to the other receptors calculated as $2^{-\Delta\Delta Ct}$. Data represent mean values, min to max range ($n = 12$, one-way ANOVA, Dunn's correction; \*\*$p < 0.01$; \*\*\*$p = 0.001$). **c** ACTH quantification in supernatants on SF cultured for 7 days without media replacement. Data are mean ± SD ($n = 15$). Dotted line represents values quantified in media without incubation with SF cells, serving as a negative control. **d** ERK-phosphorylation as analyzed by western blotting after 5 min stimulation of SF with MC agonists at the indicated concentrations. **e** Intracellular cAMP accumulation was quantified by EIA after 15 min SF cell stimulation with BMS or αMSH. The adenylyl cyclase activator forskolin (3 μM) was used as a positive control, yielding 0.58 pmol/ml. Data are mean ± SE ($n = 3$). **f** $Ca^{2+}$ mobilization in SF upon addition of αMSH or BMS (from 1 μM then serially diluted up to 10 pM), and recorded for 86 s. Ionomycin (1 μM) was used as a positive control, yielding 0.15 units of absorbance ratio at 340/380 nm. Data are mean ± SE ($n = 3$). **g** In vitro wound healing assay conducted using ibidi® chambers on SF cells stimulated with 10 μM αMSH and 1 μM BMS. Representative images show gap closure. Scale bars indicate 100 μm. Data are mean ± SE ($n = 6$, two-way ANOVA). **h** Cytokines release from SF stimulated with SAA at 10 μg/ml and treated with αMSH (30 μM) or BMS (10 μM) for 24 h, determined by ELISA. Data are mean ± SE ($n = 15$, one-way ANOVA vs. control; \*$p < 0.05$). **i** Transwell® inserts were used for SF migration (Mig) and invasion assays on Matrigel®-coated wells (Inv). Cells were treated overnight with 10 μM BMS. Data are mean ± SE ($n = 6$, Student's *t*-test vs. control). Source data are provided as Source Data file.

The selectivity of BMS was confirmed in HEK293 cells transfected with $MC_1$ (Supplementary Fig. 1J), a cell type selected because lacking MC receptors, hence devoid of this confounding element. BMS yielded a clear concentration–response curve in $MC_1$-HEK293 (calculated $EC_{50} = 2.8$ nM) monitoring cAMP accumulation. However, MC receptor can signal through multiple pathways[23]. BMS activated ERK-phosphorylation in $MC_1$-transfected HEK293 cells and displayed a partial agonistic activity at

$MC_3$-transfected HEK293 (Supplementary Fig. 1J). Given that SF expressed all MCRs, with the exception of *MC2R*, the effective contribution of $MC_1$ was further investigated. Primary SF from mice deficient in either *Mc1r* or *Mc3r* were tested for BMS pro-senescence activity. Such activity was abrogated by absence of functional $MC_1$ yet fully preserved in $Mc3r^{-/-}$ joint fibroblasts (Fig. 2h). The pro-senescence effect of BMS in murine SFs was prevented by the ERK inhibitor FR180204. Co-incubation of BMS

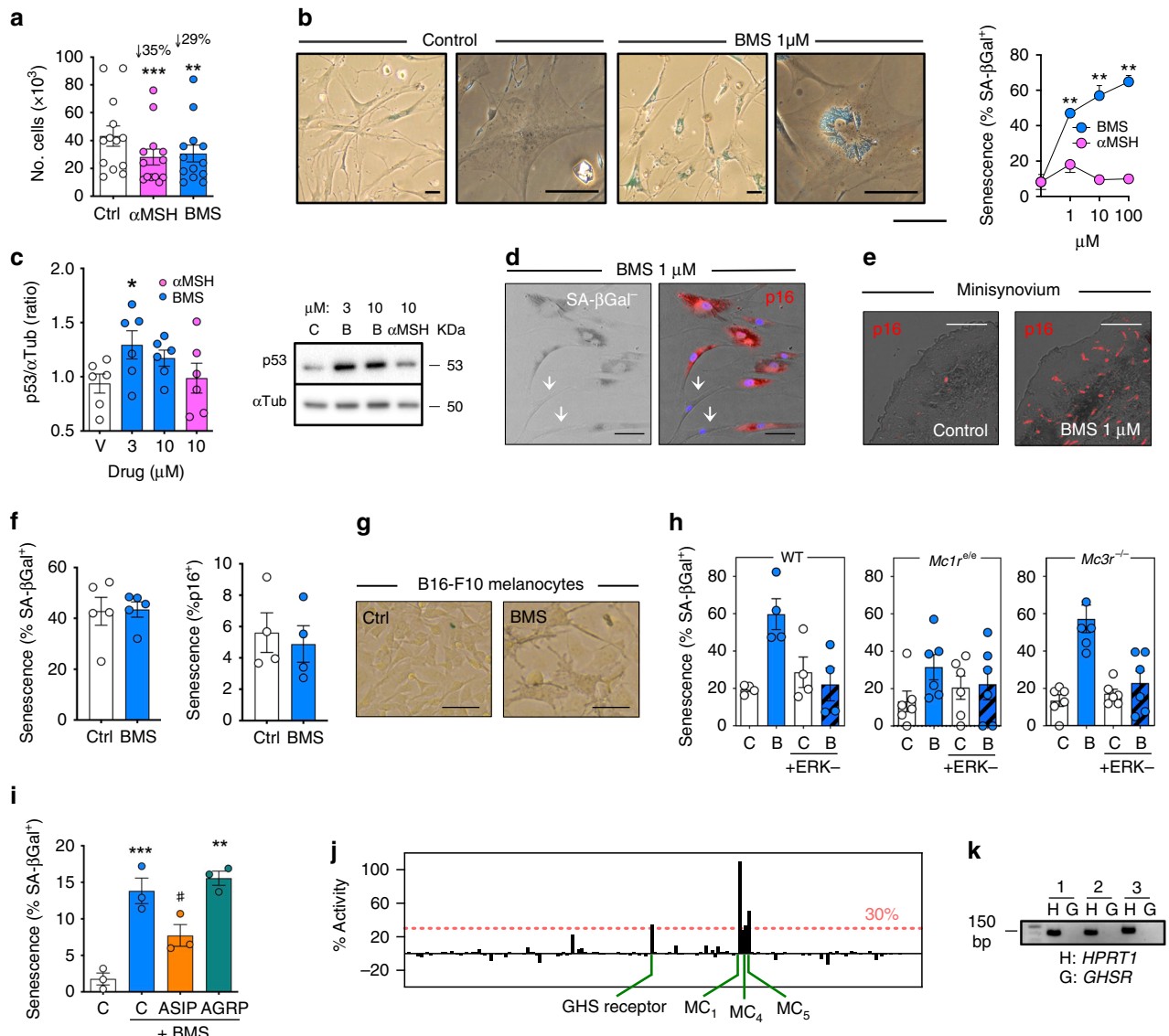

**Fig. 2 Cellular senescence induced by selective MC₁ agonism. a** SF cells were treated for 7 days with 10 μM αMSH or 1 μM BMS and proliferation assessed by cell counting. Data are mean ± SE ($n = 13$, Student's $t$-test vs. control; $**p < 0.01$, $***p < 0.001$). **b** Cells were treated with BMS or αMSH for 7 days and SA-βGal⁺ cells quantified. Scale bars indicate 100 μm. Data are mean ± SE ($n = 2$, two-way ANOVA; $**p < 0.01$). **c** p53 protein levels were analyzed by western blot after 7-day treatment of SF cells with 3 or 10 μM BMS (B) or 10 μM αMSH and bands quantified (fold change with respect to vehicle). Data are mean ± SE ($n = 6$, Student's $t$-test vs. control, C; $*p < 0.05$). **d** Immunofluorescence for p16^INK4 in SF cells treated with 1 μM BMS was performed on SA-βGal-stained cells to determine their association. Scale bars indicate 200 μm. **e** SF cells were grown on Matrigel®-based 3D spheroids for 3 weeks with or without 1 μM BMS. p16^INK4 was determined by immunofluorescence. Scale bars indicate 200 μm. **f** Effect of BMS on SA-βGal and p16^INK4 staining in human macrophages incubated with 1 μM BMS for 7 days. Data are mean ± SE ($n = 5$, Student's $t$-test vs. control, Ctrl). **g** Senescence on B16-F10 mouse melanocytes (1 μM BMS for 7 days) was determined by SA-βGal staining. Scale bars indicate 1000 μm. **h** SF were isolated from the joints of wild type (WT), $Mc1r^{e/e}$, or $Mc3r^{-/-}$ mice and treated with 1 μM BMS (B) and/or the ERK1/2 inhibitor FR180204 (ERK–; 1 μM) for 7 days. Senescence was quantified by SA-βGal staining. Data are mean ± SE ($n = 4$–$6$, Student's $t$-test vs. control, C). **i** Senescence was induced in SF with 1 μM BMS and determined by SA-βGal staining. The MC₁ antagonist ASIP or MC₃/MC₄ antagonist AGRP were used at 50 nM. Data are mean ± SE ($n = 3$, Student's $t$-test vs. control, C; $**p < 0.01$, $***p < 0.001$). **j** BMS (10 μM) was tested on the PathHunter β-Arrestin assay. Data represent % activation respect to positive controls: αMSH for melanocortin receptors and ghrelin for GHS receptor. **k** RT-PCR for the reference control *HPRT1*, and the ghrelin receptor *GHSR* on three different SF cells lines with expected bands at 130 and 148 bp, respectively. Source data are provided as Source Data file.

with the MC₁ natural antagonist ASIP resulted in decreased SF senescence, an inhibition not afforded by the MC₃/MC₄ antagonist AGRP (Fig. 2i).

Since GPCRs commonly bind multiple ligands, we applied a β-arrestin screening to monitor potential off-target effects of BMS. Out of 168 different GPCRs tested (Fig. 2j), BMS activated MC₁ as a full agonist while displayed partial activity at other MCRs; a very

weak activity (~30%) was detected for the ghrelin receptor (GHS receptor). However, RT-PCR analysis revealed that SF do not express *GHSR* (Fig. 2k). This dataset allows us to conclude that BMS is highly selective for MC₁ and its activity is not secondary to the engagement of other known GPCRs. Collectively, these data indicate that MC₁-selective activation induces cellular senescence in primary SF via phosphorylation of ERK.

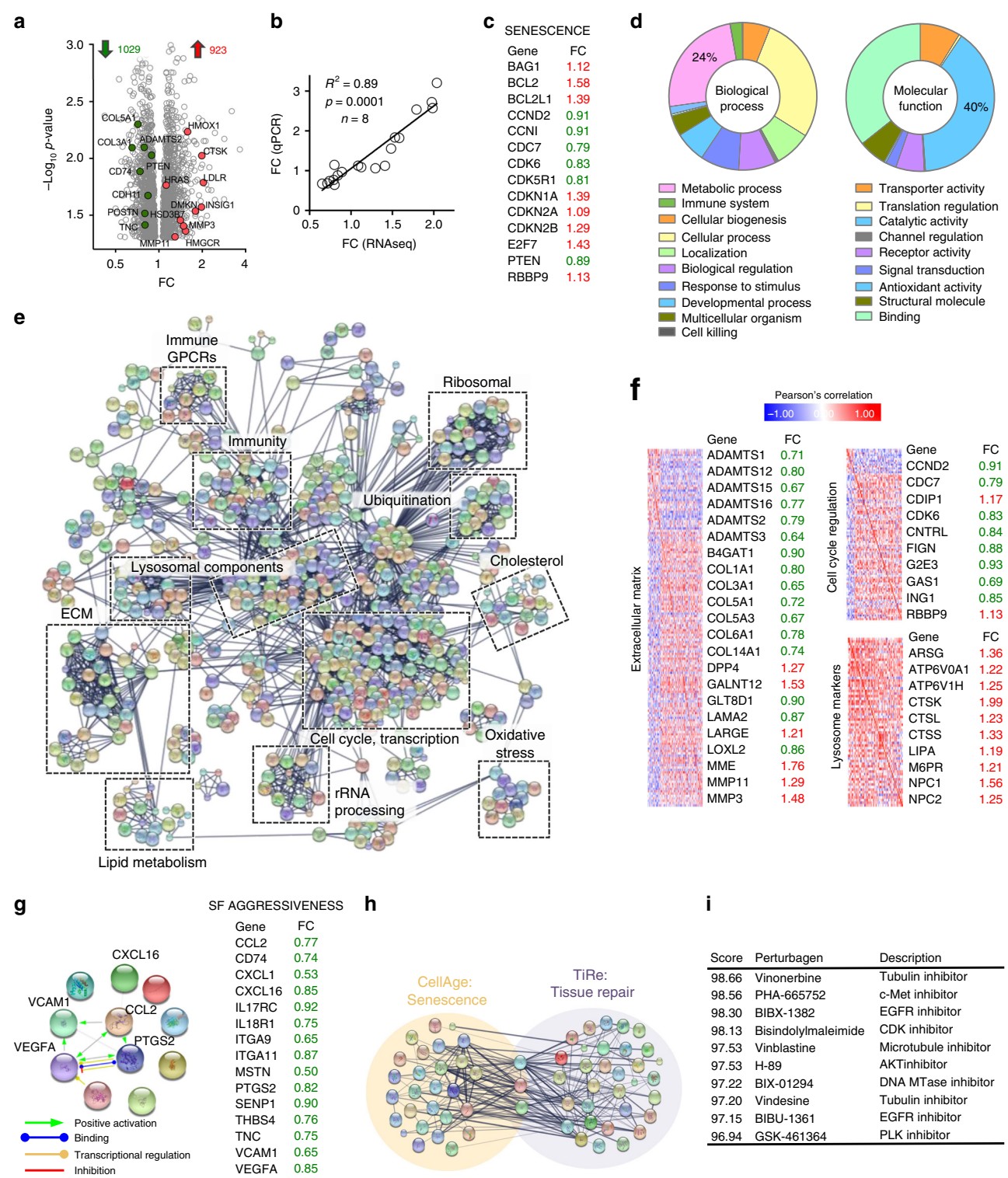

**MC$_1$-mediated senescence leads to a tissue repair profile.** Senescence is often associated with a specific secretory phenotype called SASP, which typically includes cytokines. However, the SASP induced by BMS on RA SF did not exhibit this pro-inflammatory phenotype, as levels of CCL-2, IL-6, or IL-8 were reduced (Supplementary Fig. 1K). To fully elucidate the phenotype acquired by BMS-induced senescence in SF, we conducted RNA sequencing analysis. SF were treated with BMS (using the 7-day incubation protocol) and subjected to whole-genome RNA sequencing. This

analysis identified statistically significant ($p < 0.05$) changes in 1952 genes, of which 19 were selected for validation by qPCR (Fig. 3a, b, Supplementary Tables 2 and 3), indicating a robust correlation between RNAseq and qPCR data. Modulation of several genes related to senescence and cell cycle regulation was identified (Fig. 3c, Supplementary Fig. 2A), including down-regulation of cyclins (*CCND2*, *CCNI*), up-regulation of the CDK inhibitors p16$^{INK4}$, p21, and p15 (*CDKN2A*, *CDKN1A*, *CDKN2B*), and increase of pro-survival signals typical of senescent cells (*BAG1*,

**Fig. 3 Gene expression profile acquired during MC₁-mediated cellular senescence. a** RNAseq analysis on SF ($n = 8$ patient cell lines) treated with 1 μM BMS, identified 1952 statistically significant ($p < 0.05$) differentially expressed genes which are presented as a volcano plot: fold change from BMS vs. control, against $p$ value. Highlighted genes were validated by real-time PCR. **b** Nineteen genes (details in Supplementary Table 3) were selected for expression validation using SYBR Green real-time PCR and fold changes calculated as $2^{-\Delta\Delta Ct}$. Values obtained by both techniques are presented against each other. Data are mean ± SE ($n = 8$ SF cell, Pearson's correlation). **c** Senescence-related genes significantly altered by BMS treatment and their differential expression value: up-regulated in red, down-regulated in green. **d** Functional profiling of all 1952 differentially expressed genes identified by RNAseq using Panther Classification System. **e** Protein–protein interaction (PPI) network built with all 1952 significantly altered genes using STRING. Further functional analysis was performed with DAVID and significantly enriched categories are highlighted. **f** Selected genes associated with biological functions of interest are shown with their respective expression values (up-regulated in red, down-regulated in green) together with a correlation matrix constructed with all genes in each category. **g** PPI network constructed with selected genes related to SF activation and aggressive phenotype. The fold changes of down-regulated genes (in green) induced by BMS are shown. **h** Genes related to senescence and tissue repair were retrieved from CellAge and TiRe databases, respectively, and a merged PPI network generated using STRING. **i** Connectivity map analysis shows positive association (i.e. positive score, maximum score can be 100) between the BMS-induced genes identified through our RNAseq analysis and the gene expression profiles induced by drugs included in the Connectivity Map database. Source data are provided as Source Data file.

*BCL2*). A basic analysis of bio-functions (Panther Classification System) was performed, revealing a prominent role for metabolic processes and enzymatic activity in the genes regulated by BMS (Fig. 3d). A protein–protein interaction (PPI) network was constructed with all 1952 differentially regulated genes and coupled with functional clustering using DAVID v6.8 (Fig. 3e), revealing a number of highly interacting processes altered by SF treatment with BMS. These included cell cycle regulation, metabolic processes, lysosomal components, and immune system responses. The expression of multiple extracellular matrix-related genes was dysregulated, consistent with the senescence-associated pro-repair profile as reported in different contexts[3–5]. A closer examination of these genes revealed a pro-remodeling phenotype (Fig. 3f), characterized by marked down-regulation of collagens (*COL1A1*, *COL3A1*, *COL5A1*) accompanied by increase in metalloproteases (*MMP3*, *MMP11*), and down-regulation of enzymes involved in matrix degradation (*ADAMTS1*, *ADAMTS2*). Additional genes relevant to the senescent phenotype involved in cell cycle regulation and lysosomal function were observed (Fig. 3f). The findings corroborate the senescence status of BMS-treated SF and the potential benefit that it may ensue for tissue repair. We also found down-regulation in multiple genes linked to SF aggressive phenotype, either involved in immune cell recruitment (*CCL2*), T-cell retention (*CXCL16*), angiogenesis (*VEGFA*), bone destruction (*MSTN*), or cellular activation (*CD74*, *IL18R1*) (Fig. 3g, Supplementary Fig. 2B). Out of this wealth of data we decided to focus on the relationship between senescence and tissue repair.

Query of the databases CellAge and TiRe retrieved 37 and 38 genes altered by BMS related to senescence and tissue repair, respectively. The PPI network of these genes (Fig. 3h) suggests they are highly interrelated processes. A Connectivity Map analysis revealed that BMS-induced transcriptional profile in SF is highly similar to drugs known to negatively impact on the cell cycle (Fig. 3i). Extended information on all genes reported in Fig. 3 is presented in Supplementary Table 3.

**Inactivation of Notch mediates BMS-induced senescence.** To gain more insight into the mechanism of BMS-induced senescence in SF, a pathway enrichment analysis was conducted using the whole dataset of differentially expressed 1952 genes. In line with the outcomes above, extracellular matrix and lysosome related pathways were highly enriched in BMS-treated SF (Fig. 4a), together with a strong modulation of lipid metabolism. The Notch signaling pathway emerged too, a pathway that plays a crucial role during embryonic development and cancer[26]. All genes related to Notch signaling were down-regulated by BMS treatment, including the ligands *JAG1* and *NOTCH3* (Fig. 4b, Supplementary Table 3). We confirmed the down-regulation of Notch3 protein in SF, with reduced levels of the protein evident from day 5 after

BMS treatment (Fig. 4c, d). To test the causal engagement of this pathway in BMS-induced senescence, SF were treated with BMS, in presence or absence of the delta like canonical Notch ligand 4, DLL4, and senescence quantified by p16^{INK4} staining. BMS significantly augmented p16^{INK4+} cells, but this effect was overrun by co-administration of DLL4 (Fig. 4e), indicating that inhibition of the Notch pathway is at least one of the functional mechanisms behind the pro-senescence property of BMS.

**Metabolic reprogramming contributes to MC₁- senescence.** Two of the analysis presented above indicated that changes in lipid metabolism are a prominent feature of MC₁-mediated senescence. Functional analysis of senescence-related genes (retrieved from CellAge database) revealed that metabolism is altered during senescence (Fig. 5a). Further dwelling into BMS-altered genes unveiled a marked up-regulation of the cholesterol synthesis pathway, with essentially all enzymes from acetyl-CoA till conversion into bile acids being up-regulated (Fig. 5b), with shunting off cortisol synthesis with down-regulation of *CYP11A1*. A PPI network built with CellAge genes and the cholesterol pathway genes revealed that these two processes are highly interactive (Fig. 5c). This genomic response was confirmed by elevated levels of cholesterol and extracellular bile acids upon treatment of SF with BMS (Fig. 5d).

To address whether there was a functional link between this metabolic cascade and senescence, SF were treated with BMS with or without atorvastatin to block the biosynthesis of cholesterol, by inhibiting the 3-hydroxy-3-methylglutaryl-CoA reductase (HMGCR). Atorvastatin attenuated BMS induced SF senescence by ~40% (Fig. 5e). Similarly, the increase in secreted bile acids in supernatants suggested a possible role of these lipids in the secretome characteristic of BMS-induced senescent cells. As macrophages, present in the synovial lining, express the bile acid receptor *GPBAR1* (Fig. 5f), the effect of bile acids secreted by SF on macrophage viability was determined. Human primary monocyte-differentiated macrophages were incubated with supernatants from senescent SF and the *GPBAR1* antagonist 5β-cholanic acid. Incubation of macrophages in the presence of senescent supernatants yielded a modest increase in apoptosis (Fig. 5g) that was prevented by the antagonist. The impact on macrophage apoptosis by 5β-cholanic acid was confirmed through cleaved caspase-3 immunofluorescence. Supernatants from non-senescent cells or direct exposure of macrophages to BMS did not produce any significant modulation of apoptosis (Fig. 5g, h).

**Senescence contributes to BMS pro-repair actions in vivo.** To ascertain whether senescence contributed to the anti-arthritic

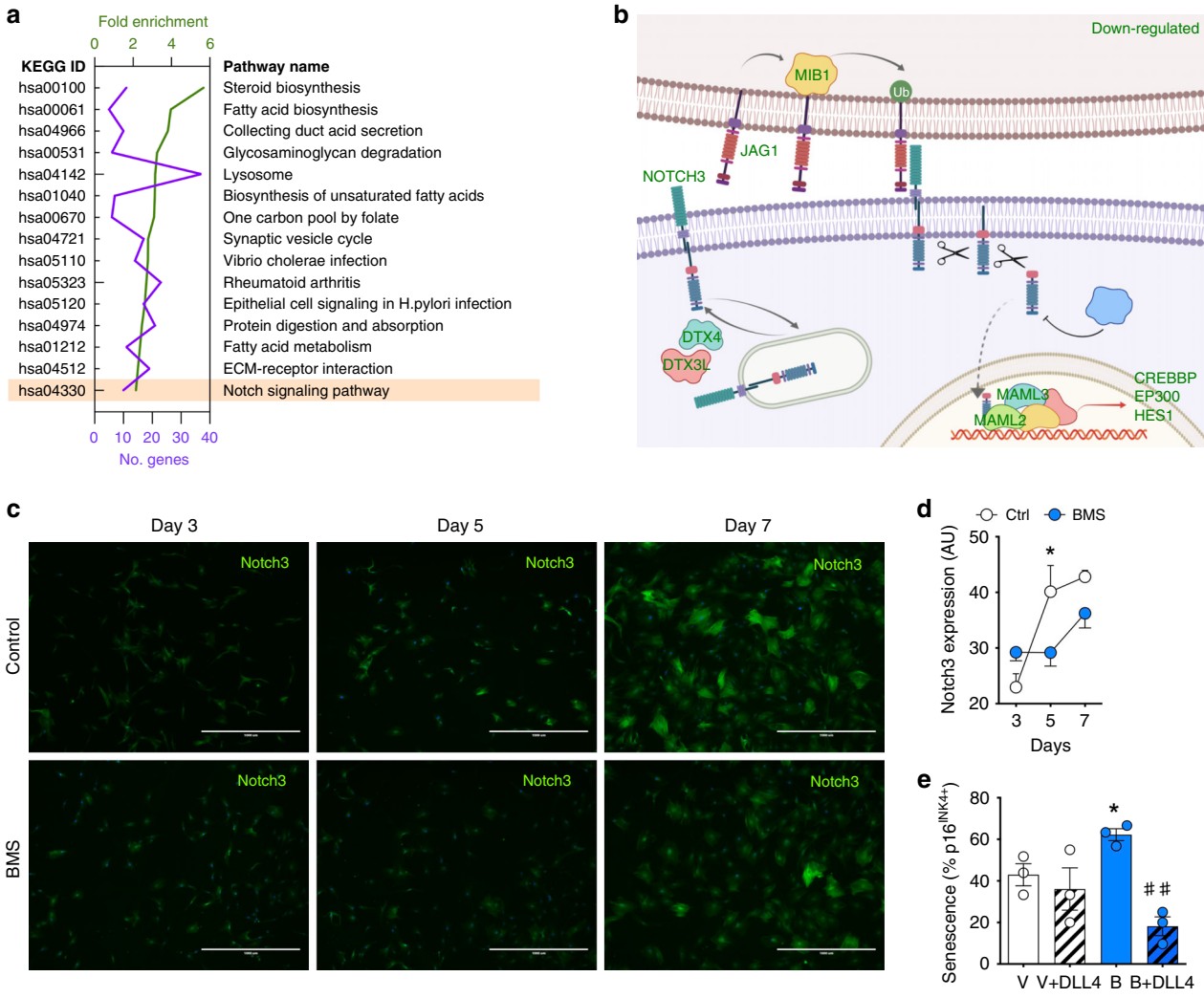

**Fig. 4 Role of the Notch pathway in SF senescence induced by BMS. a** Analysis of KEGG pathways of all 1952 differentially expressed genes following treatment with BMS (1 μM; 7 days). Number of genes annotated to each pathway (purple) and fold enrichment (green) are shown for the top 15 pathways. The Notch pathway is highlighted. **b** Specific members of the Notch pathway down-regulated in BMS-treated SF (green); figure was created using BioRender. Additional genes are shown in Supplementary Table 3. **c** Visual expression of Notch3 protein as determined by immunofluorescence on SF with or without treatment with BMS (1 μM; 7 days). Scale bars indicate 1000 μm. **d** Quantification of Notch3 protein expression on SF with or without treatment with BMS (1 μM for 7 days; $n = 3$, one-way ANOVA vs. control, Ctrl; $*p < 0.05$). **e** SFs were treated with vehicle or 1 μM BMS, in the presence or absence of the recombinant Notch ligand DLL4 (5 μg/ml) for 7 days. Senescence was measured by p16$^{INK4}$ staining by immunofluorescence and % of positive cells were quantified ($n = 3$, Student's $t$-test compared to vehicle (V*) or to BMS (B#); $*p < 0.05$, $^{\#\#}p < 0.001$). Source data are provided as Source Data file.

effect of BMS, we used a model of inflammatory arthritis, administering BMS (18 mg/kg i.p. per day) after arthritis onset (day 3) and observing inhibitory actions on clinical score and paw swelling (Fig. 6a). Positive staining for the senescence marker p16$^{INK4}$ was detected in the joint synovial lining exclusively in BMS-treated mice (Fig. 6b; validation of the p16$^{INK4}$ antibody is provided in Supplementary Fig. 4). To determine to what extent the anti-arthritic actions of BMS depended on the pro-senescence effect, we determined the effect of senolytic drugs, quercetin and dasatinib[27].

Indeed, a co-therapy protocol with senolytic drugs prevented the anti-arthritic effect of BMS with a clearer outcome on the clinical score, and less clear on the modulation of paw edema (Fig. 6c). Immunofluorescence analysis of knee sections revealed efficient removal of p16$^{INK4}$-positive cells in the animals co-treated with the senolytics (Fig. 6d). Co-staining with p16$^{INK4}$ and the SF marker cadherin-11 demonstrates co-localization and induction of senescence selectively in SF cells (Fig. 6e). Knee

and ankle joints were analyzed to measure the degree of cell infiltration and cartilage integrity (Fig. 6f, g), yielding a cumulative damage score calculated as shown in Supplementary Fig. 3. BMS reduced this damage score by ~35%, a protection prevented by co-administration of senolytics (Fig. 6f, g). Finally, treatment of mice with BMS promoted cellular senescence exclusively in SF, with no detectable induction of p16$^{INK4}$ observed in any of the seven other tissues tested (Supplementary Fig. 5).

**Association of MC1R genetic variants with SF response to BMS.** *MC1R* is a highly polymorphic gene with a high frequency of variants associated with loss-of-function, accounting for skin and hair color variation in the human population. *MC1R* variants reported in the GPCR Natural Variants Database (a total of 83) were analyzed in SF from 20 patients. Eight different variants were detected, with allele frequencies (q) ranging from 2.5% (e.g. *V60L*) to 10% (*V92M, D294H*) (Fig. 7a). No homozygotes were

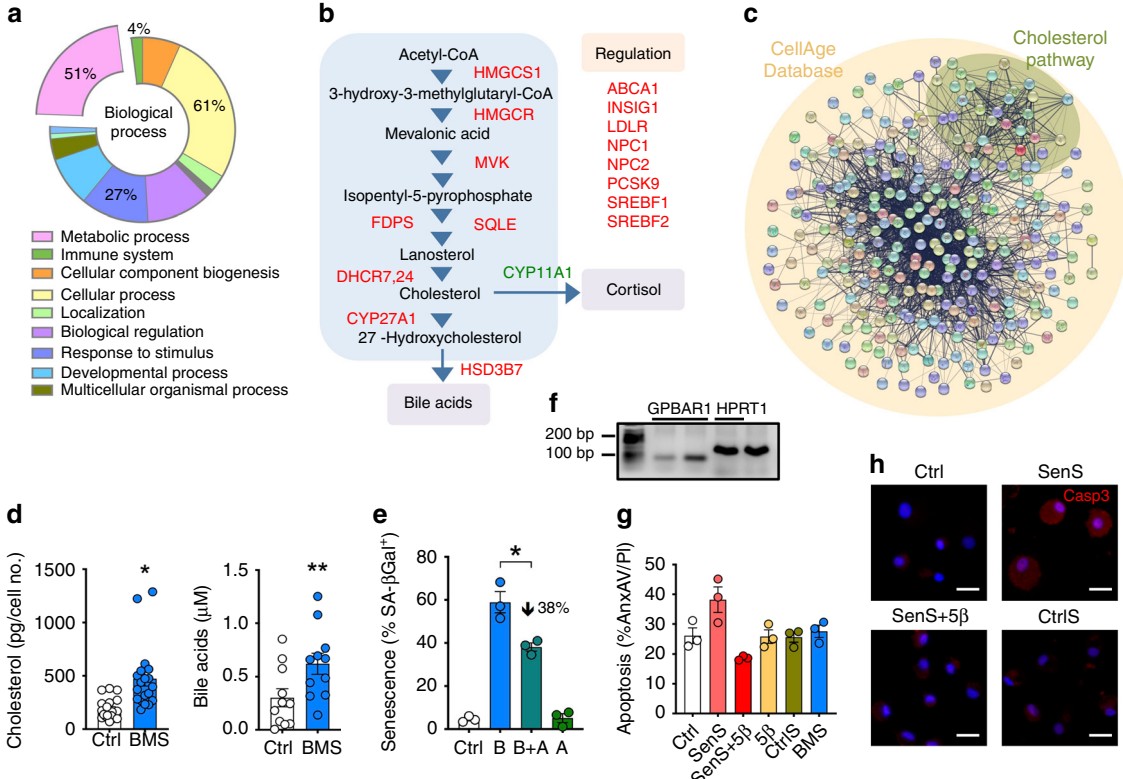

**Fig. 5 Role of cholesterol and bile acids in SF senescence. a** Functional annotation of all 279 genes included in the CellAge database. **b** Cholesterol-pathway-related genes identified by RNAseq of SF treated with 1 μM BMS for 7 days: red (up-regulated); green (down-regulated). Additional information on these genes is reported in Supplementary Table 3. **c** Protein–protein interaction (PPI) network constructed with STRING reveals the interactivity between senescence-related genes and the cholesterol pathway. **d** Levels of cholesterol and bile acids in SF treated with 1 μM BMS for 7 days were quantified by enzyme immune-assay. Data are mean ± SE ($n = 11$ for cholesterol, $n = 11$ for bile acids, Student's $t$-test vs. control, Ctrl; *$p < 0.05$, **$p < 0.01$). **e** SF were treated with 1 μM BMS (B) with or without 1 μM atorvastatin (A) for 7 days and senescence determined by SA-βGal staining. Data are mean ± SE ($n = 3$, one-way ANOVA vs. control, Ctrl; *$p < 0.05$). **f** Gene expression of *GPBAR1* and reference gene *HPRT1* on differentiated human macrophages on two different donors is shown with expected bands at 63 and 130 bp, respectively. **g** Human macrophages were incubated with supernatants from senescent SF (SenS, from SF treated with 1 μM BMS for 7 days), with SenS supernatants plus 5β-cholanic acid (SenS + 5β), supernatants from control SF (CtrlS) or directly stimulated with 1 μM BMS for 5 days. Apoptosis was assessed by flow cytometry by measuring annexin A5 and propidium iodide staining. Data are mean ± SE ($n = 3$). **h** Activated caspase-3 was assessed by fluorescent microscopy on human macrophages treated as in panel **g**. Scale bars indicate 200 μm. Source data are provided as Source Data file.

found for any of the variants. However, 55% of patients carried at least one variant, suggesting the importance of discerning their impact on $MC_1$-based therapies (Fig. 7a). Notably, 25% of patients presented with 2 different variants, yielding a total of 10 different haplotypes (Fig. 7b). A proliferation assay and senescence induction (1 μM BMS, 7 days) on cells from these 20 patients was conducted. Strikingly, all patients who did not respond to BMS (4 out of 20, hence 20%) carried the variant *D294H*, one of the loss-of-function SNPs more strongly associated with red hair[28] (Fig. 7c). Presence of this variant was not observed in the responders cells. Consistently, a robust senescence response was induced in the responders, while this effect was markedly attenuated in cells classified as non-responders and bearing the *D294H* variant ($p = 0.0002$, two-tailed Fisher's test). The fact that we did not observe abrogation of the effect is likely due to the heterozygous status and hence the presence of a wild type allele that might drive part of the effect. In line with the molecular and cellular mechanisms presented above, production of cholesterol was also markedly reduced in SF on the non-responders group. SA-βGal staining of the SF cells paralleled these results (Fig. 7c).

This genomic analysis reveals a potential therapeutic impact of variants of the *MC1R* gene like *D294H*. Other common variants

weakly associated with the red hair phenotype do not seem to impact the cellular response evoked by BMS in SF.

## Discussion

We report evidence of therapeutic induction of cellular senescence in inflammatory arthritis through selective activation of a GPCR, providing the dual benefit of arresting proliferation of hyper-proliferative aggressive fibroblasts together with the induction of a pro-reparative phenotype. Previous pro-senescence approaches, mainly developed as anti-cancer therapies[29], have been based on the use of ionizing radiation, intercalating agents causing DNA damage or small molecules to directly interfere with cell cycle components. GPCRs[30] are highly druggable targets, representing ~30% of marketed small-molecule drugs[31]. The discovery of this pro-senescence action via activation of $MC_1$ could represent a promising therapeutic avenue for arthritis and possibly other fibroblast-related conditions.

The molecular mechanisms leading to senescence are complex and affected by specific microenvironment and cellular context[1,29]. Senescence can be induced by replication exhaustion when a cell reaches the Hayflick limit and telomere attrition is sensed as an irreparable DNA damage. A second type of cellular senescence serves as a major tumor suppressor mechanism and

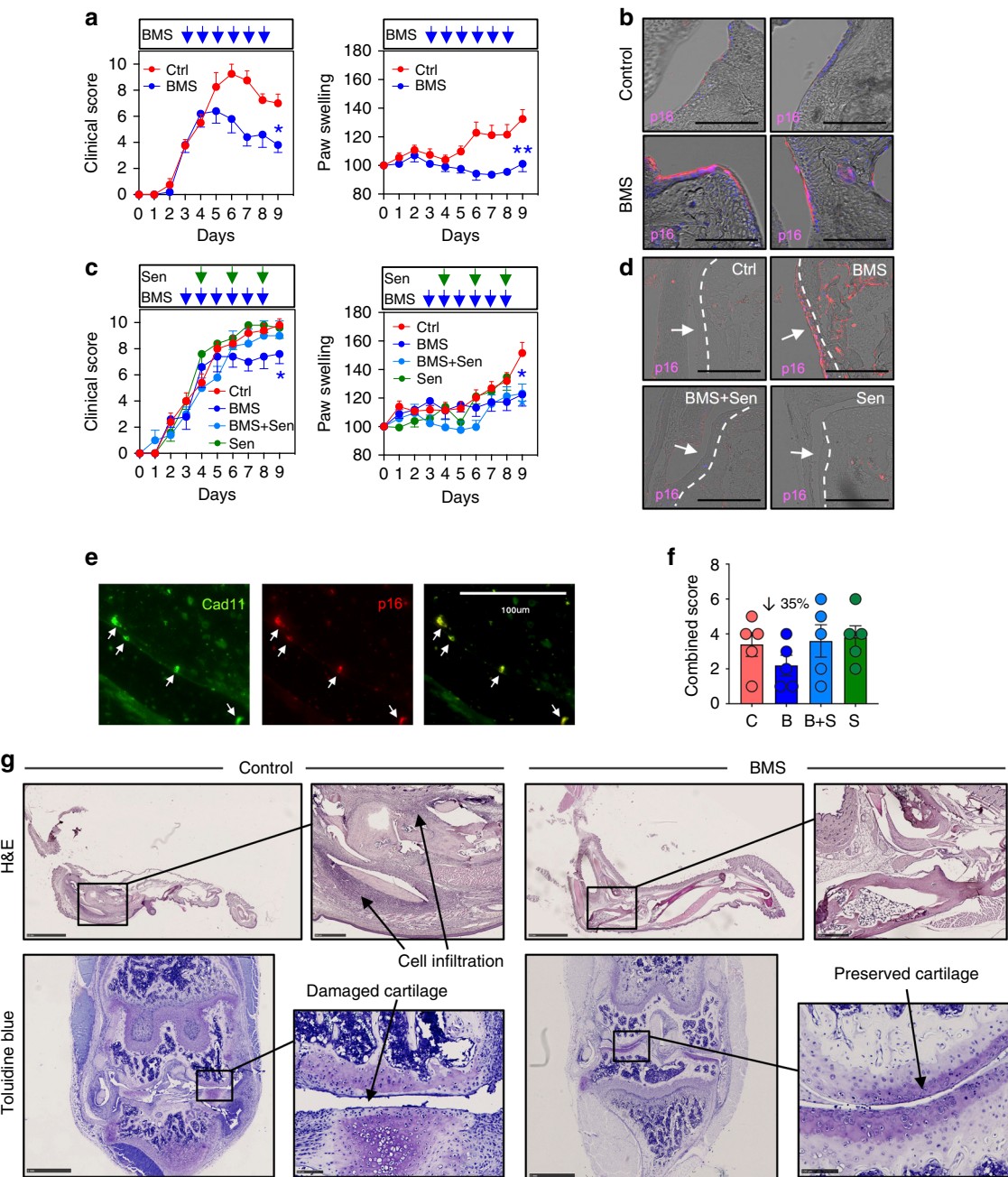

**Fig. 6 Anti-arthritic actions of BMS are associated with SF senescence. a** Arthritis was induced in C57BL/6NCrl mice with two injections of 100 μl KBN serum on day 0 and day 2. BMS was administered intraperitoneally daily at the dose of 18 mg/kg from day 3. Clinical score and paw swelling were recorded daily. Data are mean ± SE (n = 5, Student's t-test vs. control day 9, Ctrl; *p < 0.05, **p < 0.01). **b** p16INK4 expression was assessed by immunofluorescence on EDTA-decalcified and paraffin-embedded knee joint sections. p16INK4 immunostaining is shown in pink while nuclear staining with DAPI is in blue. Scale bars indicate 100 μm. **c** Arthritis was induced with two injections of 150 μl KBN serum on days 0 and 2. BMS was administered intraperitoneally at a daily dose of 18 mg/kg from day 3. From day 4, senolytics (Sen, dasatinib + quercetin at doses of 2.5 and 10 mg/kg, respectively) were administered intraperitoneally on alternate days. Data are mean ± SE (n = 5, Student's t-test vs. control at day 9, Ctrl; *p < 0.05). **d** p16INK4 expression was assessed by immunofluorescence on EDTA-decalcified and paraffin-embedded knee joints sections. p16INK4 is shown in pink. Arrows and dotted lines indicate synovial lining. **e** Co-localization by immunofluorescence of senescence marker p16INK4 and the SF marker cadherin-11, Cad11, in knee joins of arthritic mice treated with BMS. Scale bars indicate 100 μm. **f** Representative images of H&E (ankle joints) and Toluidine blue (knee joints) staining, indicating the degree of cell infiltration (H&E) and cartilage integrity (Toluidine blue) in control and BMS-treated mice. Data are mean ± SE (n = 5). **g** The extent of cell infiltration and cartilage damage were combined in a cumulative damage score (max = 6). Data values are mean ± SE (n = 5). Scale bars indicate 2.5 mm and 500 μm for H&E, and 1 mm and 100 μm for Toluidine blue. Source data are provided as Source Data file.

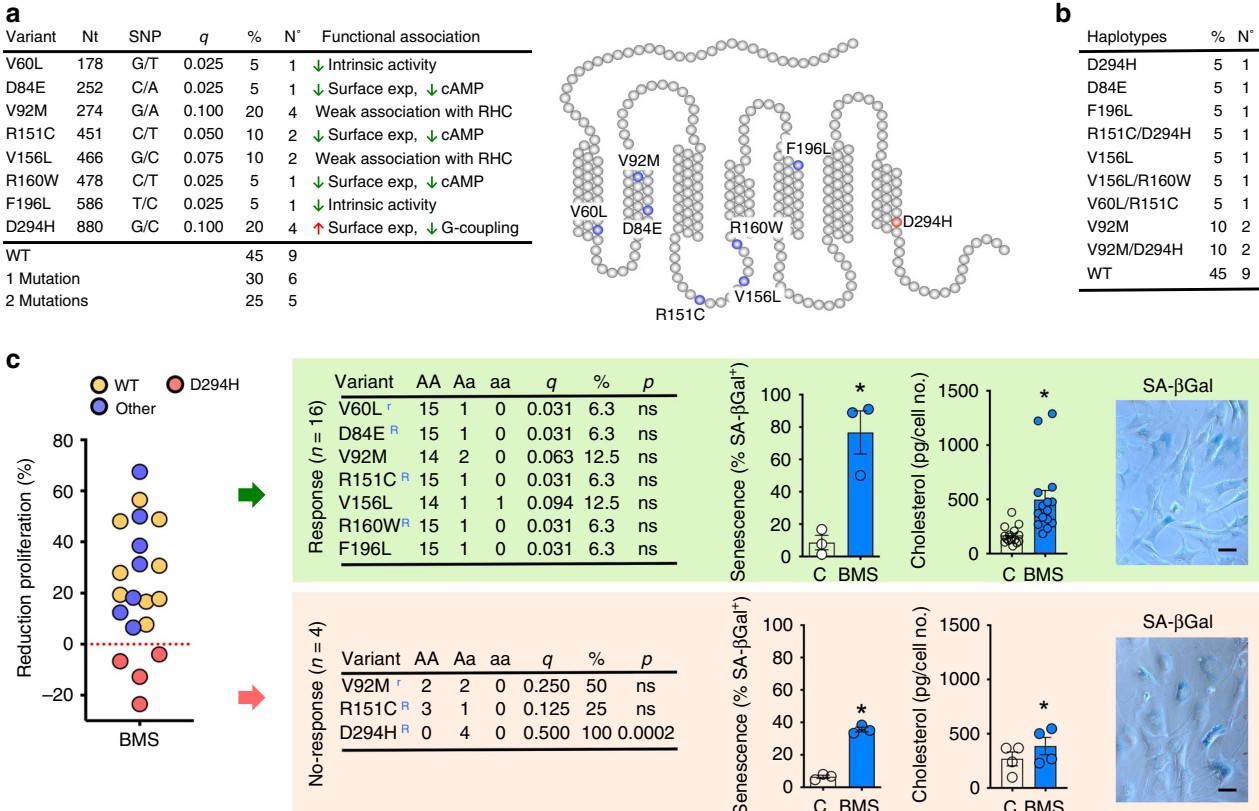

**Fig. 7 Association of *MC1R* gene variants with induction of SF senescence. a** *MC1R* variants identified in our study participants. Table includes variant name, position of nucleotide changed (Nt), specific nucleotide change (SNP), variant allele frequency (q), percentage (%), and number of patients (No.) and functional outcome described in literature for each variant ($n = 20$ patients). The location of each variant within MC$_1$ protein is shown in the scheme. **b** Percentage (%) and number (No.) of patients carrying each of the haplotypes identified (total $n = 20$). **c** Effect of BMS on proliferation of SF cell lines obtained from each of the 20 patients genotyped. Patient cells cluster into responders ($n = 16$, green panel) and non-responders ($n = 4$, yellow panel) according to the effect of BMS in the proliferation assay. For each group, the *MC1R* variant analysis is shown (A: consensus allele; a: variant allele; q: allele "a" frequency; %: proportion of allele "a" carriers; p: *p* value two-tailed Fisher's test). Moreover, superscript R: high penetrance red hair variant; superscript r: weakly associated with red hair. Quantification and images of SA-βGal staining are shown for each group (scale bars indicate 100 μm) as well as cholesterol production measured by enzyme immunoassay. Data are mean ± SE (Student's *t*-test vs. control, C, \**p* < 0.05). Source data are provided as Source Data file.

results from oncogenic stress due to activating mutations in oncogenes (e.g. *BRAF*) or inactivating mutations in tumor suppressor genes (e.g. *PTEN*). Senescence can also occur by less understood mechanisms including oxidative stress, mitotic stress or the unfolded protein response[2]. The type of senescence we report here induced by selective MC$_1$ activation is difficult to categorize into previously described classes of senescence. Following BMS treatment, SF displayed several hallmarks of senescence, including proliferation arrest, lysosomal expansion, SA-βGal and p16$^{INK4}$ staining, down-regulation in cell cycle promoters, and up-regulation in anti-apoptotic signals. It is unlikely that BMS induced replicative or oncogene-related senescence, due to absence of abnormalities like DNA-SCARS, i.e. nuclear foci indicative of DNA damage. The senescence process presented in this study might rely on mechanisms different from those previously described since they are initiated by membrane receptor signaling activation. Congruently, the SASP induced by BMS on RA SF cells did not resemble the typical pro-inflammatory profile of SASP, as levels of CCL-2, IL-6, and IL-8 were modestly down-regulated rather than up-regulated. There is evidence, however, that the nature of the SASP is context dependent and it is not always associated with a pro-inflammatory profile[32,33], a notion consistent with the existence of distinct senescence phenotypes discussed above.

Encouraged by the anti-inflammatory properties of MC agonists[23], we initiated this study to elucidate the biology of MC receptors on SF activation, observing firstly that multiple components of the MC system were expressed in SF. In receptor activation assays, SF showed a biased activation profile compared to other cell types, given that neither the natural αMSH nor the synthetic BMS (as well as ACTH and [D-Trp[8]]-γMSH, data not shown) promoted accumulation of cAMP, the canonical MC-signaling pathway[15] but their application yielded phosphorylation of ERK1/2. Although unexpected, these results are consistent with a study[34] reporting that αMSH does not increase, but rather decreases, cAMP levels in H-211 fibroblast cells. Elevation of cAMP has been reported in other types of fibroblasts[35], suggesting context-dependent outcomes and potential technical differences. The acute (24 h treatment) anti-inflammatory properties of all MC receptor agonists tested in SF were modest, with partial reduction of secreted CCL-2, CXCL-5, and IL-6 levels. However, prolonged treatment (7 days) induced cellular senescence. Of note, MC activation has not previously been related to senescence and *Mc1r$^{e/e}$* mice do not display any susceptibility to cancer (other than UVR-induced), ageing, or altered lifespan. In addition, the fact that endogenous αMSH does not induce senescence may explain why this effect has not been observed previously. Therefore, MC$_1$-induced senescence is not an effect that could be

envisaged from the physiological actions. We postulate that selective activation of $MC_1$ holds the clue to induction of senescence, leading to a functional outcome distinct to that produced by simultaneous activation of multiple receptors (attained with αMSH). This mechanism may also involve the differential formation of homo- or hetero-dimers by αMSH and BMS, leading to distinct functional outcomes. As BMS-470539 is the only $MC_1$ selective small molecule ever reported, full elucidation of this intriguing mechanism of action will require the joint effort of chemists and computational biologists towards the development of new compounds.

An omics approach was used to characterize the phenotype acquired by BMS-treated SF. Gene expression profiles observed in BMS-treated cells corroborated the senescent phenotype, with identification of several genes involved in cell cycle arrest, pro-survival signals, and up-regulation of lysosomal components, all features of senescence. Additional functional clusters were suggestive of fibroblast de-activation and promotion of tissue repair. The most prominent change includes down-regulation of collagens mirrored by an increase in MMPs expression. A marked down-regulation of ADAMTS enzymes was also observed. These enzymes are involved in the degradation of aggrecan leading to cartilage damage: down-regulation by BMS may underlie further protective actions on the RA joint.

This gene analysis revealed important clues on the molecular mechanism underpinning BMS-mediated SF senescence. Several elements of the Notch pathway were markedly down-regulated and further experimental validation suggested that this down-regulation was not consequence but rather a contributor to the senescence program activated by BMS. Notch signaling plays a crucial role during embryonic development and cancer[26], and its inhibition has been proposed as a potential anti-cancer therapy[36,37] In our hands, activation of the pathway using a recombinant ligand abrogated the pro-senescence effect of BMS. This finding can be highly relevant since the Notch pathway may play a role in the pathogenesis of RA. Activation of Notch is typical of RA synoviocytes and it mediates their proliferation in response to TNFα[38]; thus, inhibition of Notch signaling has been proposed as a viable approach to target RA[39]. Selective $MC_1$ activation may be dually beneficial in RA, via the inhibition of Notch and subsequent induction of senescence.

RNA sequencing analysis revealed a strong adaptation of metabolic programs, in particular the cholesterol pathway, with all its components (biosynthetic enzymes and regulatory proteins) being up-regulated. Cells may require an increase in the production of cholesterol for different purposes. One may be production of steroid hormones, a pathway we excluded since enzymes responsible for the conversion of cholesterol into cortisol, such as *CYP17A1, CYP11B1*, or *HSD3B1*, were not found expressed in SF (RNAseq). Neither the enzymes leading to protein prenylation using precursors produced from the HMG-CoA-reductase pathway were modified by BMS. Cholesterol is needed to provide structural integrity and fluidity to cell membranes. Its increased production in BMS-treated SF might reflect an expansion of lysosomal compartment, consistent with the up-regulation of *NPC1* and *NPC2* involved in cholesterol trafficking to lysosomes. Interactome analysis of the cholesterol pathway and senescence-related genes revealed that these two processes are highly interconnected. Congruently, inhibition of HMG-CoA reductase with atorvastatin significantly impacted on the induction of senescence by BMS. The metabolic reprogramming evoked by selective $MC_1$ activation ended to bile acid production, suggesting that augmented cholesterol might impact on cell behavior beyond lysosomal expansion.

Since SF and macrophages are the major cell types present within the synovial lining, we evaluated if SF-derived bile acids could might have any effect on macrophages, which express the bile acid receptor *GPBAR1*[40]. We observed that bile acids released by SF had mild effects on macrophage apoptosis, an action that was abrogated by application of a GPBAR1 antagonist. These data suggest that BMS-induced metabolic rewiring of SF induces senescence and impact on the type of secretome released by senescence cells. While further studies are needed to fully elucidate the role of lipid metabolism and their regulation of neighboring synovial macrophages, herein we prioritized our efforts on addressing the potential therapeutic value of SF senescence and the associated tissue repair program in the context of joint inflammation. We made use a mouse model of inflammatory arthritis[41] susceptible to MC agonists[15,22,42].

In arthritic mice, BMS induced $p16^{INK4}$ expression within the synovial lining of arthritic joints, indicating genuine induction of senescence in vivo, while at the same time, reducing clinical score and joint swelling. However, $MC_1$ receptor activation may lead to further anti-inflammatory effects via reduction of leukocyte infiltration and cytokine release[24,43,44]. Thus, we deemed important to evaluate the contribution of SF senescence on the anti-arthritic properties of BMS and we used senolytic drugs. Senolysis consists of the selective killing of senescent cells by targeting their sore point, that is, the pro-survival pathways that make them resistant to the hostile environment induced by their own SASP[2,27]. Arthritic mice treated with a seno-cocktail consisting of quercetin and dasatinib (effective in eliminating senescent cells in vivo[27]) revealed elimination of synovial lining senescent cells induced by BMS. This outcome at the cellular and tissue level was paralleled by prevention of the tissue protective actions of BMS on joint inflammation and cartilage integrity while the seno-cocktail was less effective on paw swelling (an action plausibly linked to $MC_1$ modulation of prostanoid and cytokine production).

The process of senescence might not be intrinsically detrimental, yet inefficient removal of senescent cells from tissues due to immune system deterioration associated with ageing, may yield a negative outcome[45]. Long-term studies will be required to determine if, in the context of joint inflammation, the immune system is capable of eliminating the senescent cells induced by BMS, or if a senolytic approach after a pro-senescence therapy might be required to optimize therapeutic outcome. Interestingly, a senolytic rather than a pro-senescence approach has been tested in experimental osteoarthritis (OA)[46]. The authors used the elegant trimodality-reporter mouse model p16-3MR, engineered to direct expression of pro-apoptotic proteins under the promoter of $p16^{INK4}$ gene, inducing apoptosis selectively on senescent cells[4], to reveal protection on cartilage integrity by the elimination of senescent cells. As known, there are fundamental differences between RA and OA including their relation with ageing. For OA, age is one of the most important risk factors. Senescence in chondrocytes may play an important role in OA pathogenesis by affecting cartilage integrity, similarly to other senescence-related diseases like pulmonary fibrosis in which senescence also occurs in parenchymal cells (alveolar epithelial cells). Another crucial aspect underlying the dual beneficial or detrimental outcome of senescence is the inefficient clearance of senescent cells due to impaired immune surveillance what is responsible for the damaging effects of the SASP[45]. In RA, however, the existence of aggressive SF in a proliferation mode might encourage a pro-senescence therapy. It is almost two decades since $p16^{INK4}$ activation in joints was suggested as a viable anti-arthritic therapy[47]. However, the concept of cellular senescence was not deeply considered at the time, nor was the powerful value of senescence on tissue repair. In addition, the difficulties derived from adenoviral-based transfer of $p16^{INK4}$ into the joints proposed at the time might have discouraged that approach. Here we identified a small molecule, orally bioavailable, able to induce senescence downstream GPCR

activation and without the need for gene therapy. It remains to be seen what could be the consequences, and possible side effects, associated with long-term treatment with this approach prior to further therapeutic developments.

*MC1R* is a highly polymorphic gene, and more than 200 variants have been described[48]. This variation is associated with the diversity in human normal pigmentation. *MC1R* variants are found in around 80% red/blonde hair poor tanning individuals and are strongly associated with skin cancer risk[28]. Diminished $MC_1$ function leads to impaired melanin production, which increases skin vulnerability to UV radiation and compromises nuclear excision repair mechanisms. Distinct mechanisms can lead to $MC_1$ loss of function, including uncoupling from the G-protein, limited surface expression, or alterations in the agonist binding sites[49]. Consequently, the high frequency of these variants in the population may impact on the development of $MC_1$-based therapies.

RA management is suffering from a high economical burden derived from the poor response rates to current treatments (~40% failure)[50,51]. The inability to predict treatment responses transforms RA management into trial-and-error algorithms. Precision medicine approaches should be incorporated into the drug development process to maximize treatments benefits: as such $MC_1$ is an ideal drug target in which this approach might be feasible as multiple variants and their functional outcomes are known. A high proportion (55%) of patient cells studied here carried at least one variant, and consistent with previous studies on similar populations (British/Irish), multiple highly penetrant variants (R alleles) with strong association with red hair were identified[28]. In general, *MC1R* variants did not associate with the effects of BMS on SF senescence: however, the R variant *D294H* was associated with reduced BMS-induced senescence. Although other R alleles were also identified (*R151C*, *R160W*, *D84E*), *D294H* variant considerably differs in the mechanism leading to loss of function, as it increases cell surface expression (in contrast to the other variants) possibly due to its resistance to internalization derived from defective G-protein coupling and hence deficient signaling[52]. Larger studies will be necessary to provide a conclusive association of *D294H*, and maybe other variants, with a reduced $MC_1$-mediated pro-senescence effect. The high number of possible variants and their multiple combinations as compound heterozygous, together with the absence of homozygous in this study limits the possibility of creating a rule to predict response. However, these initial findings suggest a likely role for the *D294H* variant, present in >50% of red hair individuals[28], on the efficacy of BMS to induce SF senescence, and more generally emphasizes the importance of incorporating pharmacogenomics analysis on $MC_1$-based therapies, to ensure that the right drug is given to the right patient.

In summary, selective activation of the GPCR $MC_1$ may help restoring the broken homeostasis in the RA synovium, preventing the vicious cycle of reciprocal activation between SF and macrophages and other cells within the inflamed joint environment. The molecular signature of SF upon $MC_1$-mediated senescence induction is consistent with a pro-reparative profile. We are gradually deciphering cells and mechanisms that determine resolution and appreciate persistent inflammation may derive not only from exacerbated pro-inflammatory signals but also from defective engagement of resolution programs. These data suggest that a therapeutic opportunity may arise from the exploitation of senescence to target SFs that in RA fail to switch off.

## Methods

**Human primary SFs.** Patients with RA involved in this study were diagnosed according to the 1987 American College of Rheumatology (ACR) criteria. This study complied with all relevant ethical regulations. The study was approved by the National Research Ethics Service (NRES) Committee West Midlands—The Black Country Ref. 07/H1204/191, and all participants in this study gave written, informed consent. Synovial tissue from long duration disease was collected during joint replacement surgery. Tissue from treatment-naive arthritis patients was collected by ultrasound-guided synovial biopsy, collecting tissue from multiple regions within knee, ankle, or metacarpophalangeal joints in which there was evidence of gray-scale synovitis. Ultrasound guidance was used to introduce a single portal through which tissue was sampled using custom-manufactured 2.0-mm cutting-edged forceps or a 16 g core biopsy needle (metacarpophalangeal joint). Each clinical sample originating from a patient gave rise to a cell line. Cells were grown from six to eight individual biopsies in order to overcome synovial heterogeneity; tissue samples were cut into small sections of approximately 1 $mm^3$ using a sterile scalpel. Sections were suspended in 6 ml of complete fibroblast medium and transferred into T25 flasks. Lines were incubated at 37 °C in 5% $CO_2$ and left undisturbed for a week. Subsequently, medium was changed weekly. Synovial and skin fibroblasts were cultured with complete fibroblast medium composed of RPMI-1640, 10% non-heat inactivated fetal calf serum (FCS), 1% minimum essential medium (MEM) nonessential amino acids, 100 mM sodium orthopyruvate, 2 mM glutamine, 100 U/ml penicillin, 100 µg/ml streptomycin, and used between passages 4–8. Fibroblasts were fed once weekly and maintained in culture until confluence. At feeding, 66% of the culture medium (referred to as conditioned medium) was discarded and replaced with fresh complete medium. Cells were counted using a Countess™ II FL Automated Cell Counter (ThermoFisher Scientific).

**Laboratory animals.** All animal studies were approved and performed under the guidelines of the Ethical Committee for the Use of Animals, Barts and The London School of Medicine and Home Office regulations (Guidance on the Operation of ASPA 1986) and complied with all relevant ethical regulations. C57BL/6NCrl were purchased from Charles River. *Mc1r^{e/e}*/mice were obtained from The Jackson Laboratory. *Mc3r^{-/-}* mice were originally obtained from Dr. Howard Chen (Merck Laboratories). For all in vivo studies, male, age-matched wild-type (WT) mice were used.

**Mouse primary SFs.** Hindlegs were obtained from male C57BL/6NCrl, *Mc1r^{e/e}*, and *Mc3r^{-/-}* mice. Bones were cleaned of tissue before teasing bones apart to expose synovial cavities. Tissues were enzymatically digested with 2.5 mg/ml collagenase D (Roche) and 0.25 mg/ml DNase I (Sigma-Aldrich) for 45 min, and then with 2.5 mg/ml collagenase/dispase (Roche) and 0.25 mg/ml DNase I for another 30 min. Cells were cultured in RPMI-1640 containing 10% FCS, 2 mM L-glutamine, 100 U/ml penicillin, and 100 µg/ml streptomycin. Media was changed every 3 days, and cells sub-cultured at 80–90% confluence. Cells were used at passage 5.

**Human primary macrophages.** Experiments using healthy volunteers (written consent provided) were approved by the Queen Mary Ethics of Research Committee (QMREC2014.61). Blood was collected into 3.2% sodium citrate and diluted 1:1 in RPMI-1640 before separation through a double-density gradient using Histopaque 1077/1119 (Sigma-Aldrich). Blood was layered on top of the double histopaque layer (H1119 bottom, 1077 top) and centrifuged at 1340 r.p.m. for 30 min. The peripheral blood mononuclear cell (PBMC) layer was collected and cells washed twice with phosphate-buffered saline (PBS). PBMCs were differentiated into macrophages in complete media (RPMI-1640, 10% FCS, 2 mM glutamine, 100 U/ml penicillin, 100 µg/ml streptomycin) in the presence of 50 ng/ml M-CSF (Peprotech) during 7 days.

**3D minisynovium organoids.** Cells were centrifuged and pellet resuspended in Matrigel® (Corning) at 1 × $10^6$ cells/ml. Droplets (25 µl) were plated on 2% poly-HEMA (Sigma-Aldrich)-coated plates and incubated for 1 h before addition of culture media, the same as human SF cultures above. Spheroids were cultured for 3 weeks, fixed in 4% PFA, and paraffin embedded.

**Cell transfections.** HEK293T (ATCC CRL-3216) cells were cultured in complete media (DMEM, 10% FCS, 2 mM glutamine, 100 U/ml penicillin, 100 µg/ml streptomycin) and transfected with *MC1R, MC3R, MC4R, MC5R*, or control TrueORF cDNA clones (Origene) using Lipofectamine 2000 (Invitrogen) using 2 µl Lipofectamine per 400 ng DNA in a final volume of 100 µl of OptiMEM media, added to 24-well plates containing 700 µl/well of complete media and used 24 h after transfection. Mycoplasma contamination was tested using the LookOut® Mycoplasma PCR Detection Kit (Sigma).

**PathHunter β-Arrestin assay.** A screening assay on the activation of 168 GPCR was conducted using the gpcr MAX Panel service at Eurofins Profiling Services (Fremont, CA). For agonist activity determination, cells expressing each of the receptors and carrying the Enzyme Fragment Complementation technology were treated with 10 µM of BMS for 90–180 min, and chemiluminescent signal measured 1 h after addition of detection reagent cocktail. Data are expressed as % of control ligand (αMSH for MCRs and ghrelin for GHS receptor).

**K/BxN model of inflammatory arthritis**. Arthritis was induced in male C57BL/6NCrl mice (6 weeks old) with two i.p. injections of 100 or 150 μl of K/BxN serum on days 0 and 2. Disease was monitored by assessing clinical score (score per paw: 0 = no signs of inflammation, 1 = subtle inflammation, localized, 2 = easily identified inflammation but localized, 3 = evident inflammation, not localized; max score = 12). Signs of mild bone damage was detected in some mice, but this was not consistent enough enable the monitoring of potential drug efficacy in our experimental settings. Drugs were administered i.p. on days 3–8 (BMS 18 mg/kg) and on days 4, 6, and 8 (dasatinib + quercetin at 2.5 and 10 mg/kg, respectively). This regimen was chosen to allow the induction of senescence with BMS and the rapid elimination of senescent cells with the administration of senolytics. Doses were chosen according to the route of administration and dosing schedule. To account for potential non-senolytic off-target effects, a group of mice treated only with senolytics was included. Tissues were fixed with 4% neutral-buffered formalin for 24 h, decalcified on 10% EDTA for 4 weeks (for ankles and knees), and paraffin embedded.

**Compounds**. The following compounds were used: αMSH (alpha-melanocyte stimulating hormone), BMS-470539 (1-[1-(3-methyl-L-histidyl-O-methyl-D-tyrosyl)-4-phenyl-4-piperidinyl]-1-butanone dihydrochloride) and [D-Trp[8]]-γMSH (Tocris); agouti signaling protein ASIP$_{87–132}$ and agouti-related potein AGRP$_{83–132}$ (Phoenix Pharmaceuticals); serum amyloid A (SAA; Peprotech); FR180204 (ERK1/2 inhibitor; Merck); atorvastatin, 5β-cholanic acid, dasatinib, and quercetin (Sigma-Aldrich); and DLL4 recombinant human protein (Thermo Scientific).

**Enzyme-linked immunosorbent assay and enzyme immunoassays**. The following kits were used following the manufacturer's instructions: CCL-2 and IL-6 Ready-SET-Go ELISA (eBioscience); CXCL-5 DuoSet ELISA (R&D Systems); ACTH EIA Fluorescent Kit (Phoenix Pharmaceuticals); cAMP Select EIA kit (Cayman Chemical); ERK1/2 (pT202/Y204) SimpleStep ELISA Kit (Abcam); Bile Acid Assay kit and Cholesterol Quantitation Kit (Sigma-Aldrich). The direct adenylyl cyclase activator forskolin (3 μM, TOCRIS) was used as a positive control in the cAMP assay.

**Cell viability assay**. Cells were incubated with Alamar Blue reagent (Invitrogen) diluted 1:10 in PBS for 4 h at 37 °C/5% $CO_2$. Fluorescence was measured at EX560/EM590 on a NOVOstar reader (BMG Labtech).

**$Ca^{2+}$ mobilization assay**. Cells were incubated with 2 μM Fura-2 AM (Thermo Scientific) in HBSS without $Ca^{2+}$ (Sigma-Aldrich) at 37 °C for 45 min in the dark and washed with PBS. HBSS containing 0.185 g/l $CaCl_2$ was then added prior to stimulation with agonists at the indicated concentrations, or with 1 μM Ionomycin (SIGMA) used as a positive control. Mobilization of intracellular $Ca^{2+}$ was quantified by recording the ratio of fluorescence emission at 510 nm after sequential excitation at 340/380 nm using the NOVOstar reader (BMG Labtech) for 86 s. Data correspond to time 25 s after compound addition. Data presented correspond to 25 s after drug addition.

**iBIDI™ scratch assay**. Cells were grown in two-well culture-Inserts (Ibidi) in 24-well plates in complete media (RPMI-1640, 10% non-heat inactivated FCS, 1% MEM nonessential amino acids, 100 mM sodium orthopyruvate and 2 mM glutamine, 100 U/ml penicillin, 100 μg/ml streptomycin) until confluent. Inserts were removed and cells washed prior stimulation with test compounds. Images were acquired at times 0, 24, and 48 h. Gap size was quantified with ImageJ.

**Cell migration assay**. Cells were seeded in SF medium, as above, but without serum on Transwell inserts (6.5 mm diameter, 8 μm pore size; Corning) coated with Matrigel® Basement membrane matrix (Corning) for invasion assays, and non-coated for migration assays. Ten percent FCS media was added to outer compartment. Cells were incubated overnight, stained with 1% crystal violet (Sigma-Aldrich), and counted.

**Apoptosis assay**. Cells were stained with FITC-AnxAV and propidium iodide (PI) using the Apoptosis Detection Kit I (BD Pharmingen), by staining samples with 5 μl of Annexin A5 and 10 μl of PI for 15 min at room temperature and analyzed by flow cytometry (BD FACSCalibur™).

**Western blotting analysis**. Cells were lysed using RIPA buffer (Thermo Scientific) containing Protease Inhibitors Cocktail at 1:100 dilution (Calbiochem). Samples were subjected to standard SDS-PAGE, and transferred onto PVDF membranes (Merck Millipore). Antibodies and dilutions used were: anti-phospho-ERK1/2 (1:1000; Cell Signaling), anti-ERK1/2 (1:1000; Cell Signaling), anti-α-Tubulin (1:5000; Sigma-Aldrich), rabbit anti-p53 (1:2000; Abcam), HRP-conjugated anti-rabbit and anti-mouse IgG (1:2000; Dako). Bands were quantified using ImageJ.

**Immunofluorescence analyses**. Cells were fixed with 4% PFA and permeabilized with 0.3% Triton X-100. For mouse tissues, antigen retrieval was performed by incubating slides in basic buffer (10 mM Tris base, 1 mM EDTA, 0.05% Tween 20, pH 9.0), for 5 min at 95 °C. Samples were incubated overnight with primary antibodies: anti-human/mouse p16[INK4a] (1:100; Abcam), anti-mouse p16[INK4] (undiluted; CNIO, clone PABLO33B), anti-cadherin-11 (1:10; Cell Signaling), anti-cleaved caspase-3 (1:100; Cell Signaling), anti-Notch3 (1:200; Abcam). Secondary antibodies (1:100; Thermo Scientific) were incubated for 2 h: anti-mouse and anti-rabbit IgG Alexa Fluor 594, anti-rabbit IgG Alexa Fluor 647, anti-rat IgG Alexa Fluor 647. Slides were coverslip with ProLong Diamond Antifade Mountant with DAPI (Thermo Scientific) and images acquired at ×20 magnification on the EVOS FL Imaging System (Thermo Scientific). Exposure and sharpness were kept constant for all images for comparison.

**Cell and tissue staining**. The following kits (Abcam) were used as per the manufacturer's instructions: Senescence Detection Kit (SA-βGal) and Fontana-Masson (melanin). H&E staining was performed on de-paraffinized and rehydrated sections using hematoxylin and eosin (Sigma-Aldrich) and mounted in Entellan mounting medium. Cell infiltration was scored as: 0 = not detected, 1 = mild, 2 = moderate, 3 = severe. Cartilage staining was performed with toluidine blue (1%, 10 min) (Sigma-Aldrich). Cartilage integrity (loss of sulfated proteoglycans staining) was scored as: 0 = normal, 1 = mild loss, 2 = moderate loss, 3 = total loss of staining. Images were acquired at ×20 magnification on a Nanozoomer S210 Slide Scanner (Hamamatsu). Exposure and sharpness were kept constant in all images for comparison.

**Melanin measurement**. B16-F10 cells were cultured in 10% FCS αMEM. Cells were treated for 4 days and stained with Fontana-Masson staining or centrifuged to visualize melanin in cell pellets. Melanin content in supernatants was determined by measuring absorbance at 405 nm.

**Gene expression analyses**. RNA was extracted using Direct-zol RNA MiniPrep with in-column DNase I digestion (Zymo Research). cDNA was synthesized (1 μg RNA) with SuperScript VILO MasterMix (Invitrogen). End-point PCR was performed with ReddyMix PCR MasterMix (Thermo Scientific). Real time-PCR was performed with Power SYBR Green PCR MasterMix (Applied Biosystems) on the ABI Prism 7900HT Sequence Detection System. Expression was calculated as $2^{-\Delta\Delta Ct}$ [53] using HPRT as a reference gene[54]. Quantitect primers (QIAGEN) used are shown in Table S2, with an annealing temperature of 55 °C used for all primers. RNA sequencing was conducted at Eurofins-GATC services (Konstanz, Germany) using non-stranded random primed cDNA on an Illumina HiSeq Genome Sequencer, sequence mode HSHOv4 SR50 (HiSeq rapid run, 50 bp single read). Sequencing reads were aligned to the human genome build hg38/GRCh38 with the HISAT2 aligner. Transcript quantification was performed with htseq-count (HTSeq package v0.6.1p1), using GENCODE v23 human gene annotation (Ensembl release 81). The read count data were filtered to keep genes that achieve at least one read count per million (cpm) in at least four samples. Hence, to be considered expressed, any given gene needs to reach a value of at least cmp = 1 in at least the 25% of samples (i.e. 4 samples). Reads per kilobase per million mapped reads (RPKM) values were calculated with the conditional quantile normalization (cqn) counting for gene length and GC content in the R statistical environment via Bioconductor. Genes with $p > 0.05$ were included yielding a working list of 1952 differentially expressed genes. This research utilized Apocrita HPC facility supported by QMUL research-IT. Data were deposited in GEO: access code GSE98658.

**Functional analysis and databases**. The functional analysis of the previously identified differentially expressed genes (1952) was conducted with DAVID v6.8 and Panther Classification System v12.0 by uploading the full set of genes. The gene expression profile induced by BMS was used to query the Connectivity Map v02 perturbation-driven gene expression dataset using default settings. PPI networks were constructed using STRING v10.5 by uploading the full gene list and excluding disconnected nodes from the resulting network. We also used information from the following databases: CellAge Database v[beta], TiRe Database 1.0, and GPCR NaVa Database.

**MC1R gene sequencing**. The entire coding region of *MC1R* was amplified using the primers Forward: 5′-GAAGAACTGTGGGGACCTGG-3′ and Reverse: 5′-GGGTCACACAGGAACCAGAC-3′. PCR products were purified with QIAquick Gel Extraction Kit (QIAGEN) and subjected to Sanger sequencing at Eurofins-GATC services (Konstanz, Germany) using an ABI 3730xl DNA Analyzer and basecaller KB v1.4.1.8. Electropherograms were analyzed using *.ab1 files and variants detected using SnapGene® Viewer v3.0.3 and the Homo sapiens *MC1R* reference sequence NM_002386.3.

**Statistical analysis**. In each experiment included in this study, the value of *n* is always defined as number of biological replicates (e.g. participants, mice) and not technical replicates. Statistical parameters including the exact *n* value for each experiment, nature of data shown (mean ± SE or mean ± SD), and statistical significance are reported in the figure legends. Data are considered statistically significant when $p < 0.05$. One-tail Student *t*-test, one-way ANOVA, or two-way

ANOVA were used as appropriate. In the figures, asterisks or hashes denote statistical significance ($*,\#p < 0.05$; $**,\#\#p < 0.01$; $***p < 0.001$). Statistical analysis was performed in GraphPad PRISM v7.

**Reporting summary**. Further information on research design is available in the Nature Research Reporting Summary linked to this article.

## Data availability

All data supporting the results presented herein are available from the corresponding authors upon reasonable request. Raw data files for the RNA sequencing analysis have been deposited in the NCBI Gene Expression Omnibus under accession number GEO: GSE98658. The source data underlying graphs and un-cropped gels and blots in the main figures and Supplementary Information are provided as a Source Data file.

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

## Acknowledgements

This research was funded by Medical Research Council (grant MR/K013068/1) and William Harvey Research Foundation. T.M.-M. was supported by Medical Research Council (grant MR/K013068/1) and Versus Arthritis UK (grant 21274). The work of the authors C.D.B. and A.F. is supported by NIHR Oxford and Birmingham Biomedical Research Centres. The views expressed are those of the authors and not necessarily those of the NHS, the NIHR, or the Department of Health and Social Care. We thank J.D. Turner for technical assistance. We thank Dr. M. Serrano, K. Meyer, M. Garcia (Institute for Research in Biomedicine, Spain), and G. Roncador (Spanish National Cancer Research Centre, Spain) for kindly providing and assisting with the anti-p16[INK4] antibody, and Dr. S. Godinho and J. Simon (Queen Mary University of London, UK) for providing mouse cell lines.

## Author contributions

M.P. and T.M.-M. conceived and designed the study, acquired funding, and wrote the manuscript. T.M.-M. designed and performed all experimental part. T.M.-M., A.N. and C.C. analyzed data. T.M.-M., M.P. and C.D.B. interpreted data. A.F. provided human cells from RA patients. M.P. supervised the study. All authors revised and approved the manuscript.

## Competing interests

The authors declare no competing interests.
