## [Peer Review File · Nature Communications]

Reviewers' Comments:

Reviewer #1:

Remarks to the Author:

The manuscript by Montero-Melendez et al appears to show that melanocortin receptor 1 activation by the drug BMS (but not the endogenous agonist α MSH) in synovial fibroblasts from patients with rheumatoid arthritis induces cell senescence, accompanied by changes in the transcriptome, and that such senescence serves beneficial and pro-repair role in the joint of mice with induced RA.

The paper is a wide-ranging but strange mix of some good science, some much weaker studies, and the text itself shows huge variability in quality and coherency - the Results and Figure legends include a large and confusing number of undefined abbreviations, with no clear rationale behind the design of the studies, while the Discussion is clear and makes a compelling argument. Overall, the range of experiments conducted and the congruence between data from different models is supportive of the overall conclusion that BMS activation of MC1R may be beneficial in RA, but the science needs some serious tightening, including fundamental issues such as inclusion of appropriate controls at all times and care with interpretation of -omics data sets, as well as making the paper more reader-friendly by setting out a robust scientific rationale for each set of experiments and explaining the data more clearly (both in the main text and in Figure legends and figure labelling).

The initial rationale for studying melanocortin signalling and the major scientific question to be addressed are not at all clear from the experimental results section. Figures legends are too brief and too littered with unexplained abbreviations to be helpful to the reader. There are many omissions and inconsistencies: eg Fig 1A, B data show gene expression (line 103 states mRNA) but the legend states end-point PCR and qPCR without mentioning that the starting material was RNA - such information lies buried in the Methods but is required in the figure legend for correct interpretation of the data. Lanes 1-4 of Fig 1A are not described anywhere. The fold change calculations in Fig1B need further explanation - a delta CT of 12 and one of 16 are said to represent a 10 fold difference in gene expression - how is this calculated? Normalisation against the factor being measured is not usual practice - if normalisation is against HPRT, then another unchanging housekeeping gene should also be used: a large number of genes show altered expression on senescence so any hint of a senescent phenotype should require use for at least two normalisation standards in reverse transcription-qPCR. Quantification eg of p53 levels in Fig 2C does not seem consistent with the gel shown.

The use of forskolin is not explained in the legend/accompanying text and it is left to the reader to piece together the rationale - similarly, the use of the ionophore is not explained, and the numerical value given on the graph does not correlate with the concentration stated in the Methods section. There are errors of this type throughout, as well as odd phrasing and grammar, omission of definite articles in parts, incorrect use of singular instead of plural in occasion (e.g. line 559 'receptor'), and use of 'on' instead of 'of' in several parts. Such lack of care in proof reading does not inspire confidence in the care taken to conduct rigorous experiments, particularly when labelling of figures is missing (e.g. scale bars), units potentially incorrect (e.g. IL-6 is measured in μ g/ml which is extremely high - senescent fibroblasts usually secrete in the order of ng/ml or pg/ml of IL-6 into the culture medium). Parts of figures are incorrectly cross-referenced (e.g. line 883 - I presume G is meant, not F?) Use of 'and' instead of 'or' is deeply misleading (e.g. line 745, but multiple such cases throughout). Use of dashes instead of parentheses is also confusing especially when used with numerical values e.g. dasatinib dose line 844. Text and figure labelling are not always consistent eg CtrlS in fig and ConS in text. All gels require labelling of size markers.

Measuring senescence-associated beta galactosidase (SABG) in cell types such as macrophages cannot be used for assessment of senescence - their high lysosomal content gives a high baseline of SABG staining (as the authors observe in Fig 1F but do not comment on). The high baseline rate

of macrophage apoptosis also calls into question the validity of the method used - are the cells excessively stressed? Since there is currently no single specific biomarker for senescence, the authors have sensibly examined a number of parameters including a proxy for proliferation rate (cell number, though method of assessment is not given even in methods), SABG, and p16, as well as looking at morphology/size for SFs. They later rely on single markers to determine senescence (either SABG or p16), neither of which is sufficient alone to show senescence: SABG is simply a marker of lysosomal stress, whereas p16 is elevated on cell cycle arrest, either transient or permanent, and is not always present in senescent cells (it is lineage specific).

Figure labelling is inconsistent, as are treatments e.g. various doses of BMS are employed even within a single figure, but without any justification. There are occasional oddities for which the logic is not presented e.g. why are adipose and liver cells included at the end of Fig 5?

Perhaps my greatest concern lies in interpretation of the RNAseq data. A major premise of the paper is based on BMS-dependent changes in gene expression, with extensive bioinformatics work up (with a number of nice-looking but not necessarily deeply informative figures - though I appreciate a number of -omics papers are similar). Much is made in the paper of 'marked' changes, 'major' upregulation or dysregulation, 'significantly altered' (line 798) but in all cases, the genes in question show fold changes of >0.5 and <2 i.e. not normally considered significant. Biologically, such changes may be important but statistically they are not: if a large number of factors in a pathway all change in the same direction in response to treatment, then that may be significant - proper statistical analysis of the pathways needs to be conducted. Fold changes of the cholesterol pathway are not even given. In other tests (eg t test), it is necessary to state whether one or 2 tailed tests were used. In many instances (measurement of levels of cytokines, cholesterol, bile acids etc etc), box and whisker plots or violin plots would be much more informative than bar graphs. Since MMPs and ADAMs both remodel the ECM, how do the authors account for opposite changes in expression on BMS treatment being consistent with senescence and tissue repair?

Where a patient cohort of 20 individuals is used, it is not helpful to state that 5% show a particular haplotype - that represents just one individual. Instead, the number of patients from the cohort should be given as with this sample size, % is misleading. On a further point of statistics, it appears odd that the symptom duration standard deviation is far greater than the mean - it is really the case that there is a ± 10 year SD in a cohort who have suffered symptoms for ~ 6 years? That implies some patients have been included who have suffered for a large negative time period?

The Methods section is written in too brief a manner to allow other labs to repeat the studies e.g. antibody dilutions are given but not clone number or other unique identifier; % are given where molarities are required (eg glutamine in culture medium).

Image processing states that exposure was optimised for printing - but it is critical that gain and exposure times are kept constant for immunofluorescence to allow direct comparison between samples and controls.

While the Discussion mentions improvements in OA on senolytic therapy (ie the opposite of findings present here in RA), it could benefit from more balanced presentation of where senescence may be beneficial and where it is detrimental - eg idiopathic lung fibrosis is senescence-associated and senolytic treatment appears helpful. Acute wound healing needs acute senescence while chronic senescence blocks wound healing.

Overall, I think there may be real value in this work, and the authors may have come upon a very exciting potential new therapy for RA with an additional avenue for patient stratification and hence precision medicine in RA, but the manuscript and figures need major tidying up, tightening and

improving in scientific precision before the paper is suitable for publication.

Reviewer #2:

Remarks to the Author:

Montero-Melendez and co-workers studied the melanocortin (MC) in rheumatoid arthritis in humans and animals. They showed for the first time that MC type 1 receptor (MC1) are need to induce a cellular senescence phenotype characterized by arrested proliferation, metabolic re-programming and marked gene alteration resembling the remodeling phase of wound healing, with increased MMP expression and reduction in collagen production. This could be a novel ways to control joint inflammation and arthritis.

This study is very interesting, experiments are well planned, methods and statistical analysis seem appropriate, results are clear and the references are updated.

I have the following minor comments:

1. The paper should be shorter
2. Please explain better GPCR?
3. Introduction: authors should be quote other important effects of melanocortins (e.g.: protective effect of MC on circulatory shock, myocardial and cerebral ischemia, experimental Alzheimer disease) and respective groups of research.

Reviewer #3:

Remarks to the Author:

In this manuscript, Montero-Melendez et al. reported a novel finding that activation of the melanocortin type 1 receptor (MC1) by a compound named BMS-470539 (BMS) can induce a senescence phenotype in synovial fibroblasts. This senescence phenotype includes proliferation arrest, lysosomal expansion, SA-bGal and p16INK4 staining and genome-wide gene expression changes. Furthermore, the authors tested the functional importance of this RF senescence induced by BMS in a K/BN arthritis model. The manuscript also discussed the effect of MC1R variants on BMS-induced RF senescence. Overall, the main finding (BMS induces a senescence phenotype in synovial fibroblasts) is novel and interesting. However, with the consideration of the below major points, the results are mostly descriptive and preliminary without much mechanisms revealed. Importantly, the data quality in the arthritis model is poor, and the results did not fully support the authors' interpretation. Further, some important outcomes from senescence, such as SASP (senescence induced secreted factors, cytokines) and inflammation, are not examined in this study. It is therefore not clear for the long term effects of BMS-induced RF senescence on RA.

Major concerns:

1. One critical feature associated with senescence is SASP (senescence induced secretome, for examples growth factors and cytokines). The secreted factors and cytokines often trigger immune response (both innate and adaptive) and chronic inflammation, which may deteriorate RA inflammation. However, the authors did not test this important feature, so it is hard to judge the therapeutic importance of long term BMS effects on RA.
2. The off-targeting effects of BMS in vivo: although the authors did not find BMS induced senescence in liver and adipose in the arthritis model, did they examine other tissues or organs, such as kidney, spleen, brain, lung and skin? Because they found BMS can induce senescence in human dermal fibroblasts, the senescengenic effect of BMS seems broad not limited to synovial fibroblasts.
3. Although the group performed profound bioinformatic analysis, there is little data to experimentally elucidate the mechanisms suggested by these bioinformatic information.
4. The bile acid secreted by SF induced macrophage apoptosis. With consideration of the importance of macrophages in RA, does this macrophage apoptosis affect inflammation and immune states of the arthritis mice?
5. Since BMS activates MC1, have the authors tested the general effect of MC1 activation in brain

function--the food intake regulation, in vivo in the arthritis model?

6. The evidence of lysosomal compartment is not clear.

7. Have the authors tested the apoptosis of SF in response to BMS?

8. Many experiments lack appropriate controls. For example, Fig. 1C should have controls, including skin fibroblasts and OA fibroblasts. Fig 1E, no positive controls provided.

9. Some data quality is poor: the western blot bands in Fig 1D are blurry. The quality of histological slices in Fig 5 F and G is poor; it is hard to tell the details of the structures and generally the 4x pictures should be provided too. The locations in F seem different between conditions, and tissues lost integrity. The criteria of synovitis, such as immune cell infiltration, panus formation extent and tissue destruction (cartilage and bone), was not clear although the scores 0 (no synovitis), 1 (mild) and 2 (moderate) were mentioned. The BMS group in G shows more bone destruction, which is against the interpretation of the data. Some pictures show more joints, while the others only show surface of the joints. The tissue locations shown in different conditions are not comparable.

10. Some experiments or results lack critical information to understand: How was BMS administrated in vivo (orally, ip, iv)? What are those "other members of the MC pathway" in Fig 1A? In Fig 4D, what are the specific EIA? Suppl Fig 1, what are the details of the cell culture?

11. The information about the synovial fibroblasts used in this manuscript is confusing. Are these all from the 20 RA patients? or additional patients included? Some experiments used $n < 20$ (for example, in Fig 1A, B, C. In Fig 3, $n = 8$ donors, what are these 8 donors?), so how did the authors select the SF cells for different experiments?

12. The introduction did not provide necessary background of previous studies on melanocortin system, senescence, and their relation with health and diseases, in particular with inflammation and arthritis (both RA and OA), especially the recent article published in Nature medicine regarding local clearance of senescent cells that attenuates OA, which provides proof-of-concept evidence of the importance of clearing senescent cells in OA.

Minor points:

1. Details of RNAseq analysis are missing. It is insufficient to only list the names of bioinformatics resources.

2. It should be helpful to understand the dataset if the authors could describe results more clearly and provide more details.

Rebuttal Letter

Reviewer#1

The manuscript by Montero-Melendez et al appears to show that melanocortin receptor 1 activation by the drug BMS (but not the endogenous agonist α MSH) in synovial fibroblasts from patients with rheumatoid arthritis induces cell senescence, accompanied by changes in the transcriptome, and that such senescence serves beneficial and pro-repair role in the joint of mice with induced RA.

The paper is a wide-ranging but strange mix of some good science, some much weaker studies, and the text itself shows huge variability in quality and coherency - the Results and Figure legends include a large and confusing number of undefined abbreviations, with no clear rationale behind the design of the studies, while the Discussion is clear and makes a compelling argument. Overall, the range of experiments conducted and the congruence between data from different models is supportive of the overall conclusion that BMS activation of MC1R may be beneficial in RA, but the science needs some serious tightening, including fundamental issues such as inclusion of appropriate controls at all times and care with interpretation of -omics data sets, as well as making the paper more reader-friendly by setting out a robust scientific rationale for each set of experiments and explaining the data more clearly (both in the main text and in Figure legends and figure labelling).

We thank this Reviewer for appreciating the scientific value of our work and for being supportive for the overall conclusion of our study. We also appreciate the very thorough revision and the time spent by this Reviewer on our manuscript and we thank for identifying several editing aspects in the text, figures and legends: we agree they needed to be tighten up to help readers receiving the correct interpretation of our work, and have done so. Moreover, specific points are addressed below:

The initial rationale for studying melanocortin signalling and the major scientific question to be addressed are not at all clear from the experimental results section.

We have expanded the last paragraph on the Introduction (page 5) to clarify the rationale of our study. We also extended on the therapeutic applications of melanocortin drugs, as requested by Reviewer #2.

Figures legends are too brief and too littered with unexplained abbreviations to be helpful to the reader.

We have revised all Figure legends and we have substantially expanded the details (considering the limit of ≤ 350 words required by Nature Communications).

There are many omission and inconsistencies: eg Fig 1A, B data show gene expression (line 103 states mRNA) but the legend states end-point PCR and qPCR without mentioning that the starting material was RNA – such information lies buried in the Methods but is required in the figure legend for correct interpretation of the data.

We have changed the Figure 1A legend to indicate that the gene expression analysis performed by PCR were conducted using RNA.

Lanes 1-4 of Fig 1A are not described anywhere. The fold change calculations in Fig1B need further explanation – a delta CT of 12 and one of 16 are said to represent a 10-fold difference in gene expression – how is this calculated?

Figure 1A shows the PCR results for 4 patients (“n=4” shown in the legend), each line representing one of them. We have now specified in Figure 1A legend that each line represents one SF cell line obtained from a patient out of the n=4 in total.

Fold changes are calculated with the widespread and accepted method created in 2001, using the formula of $2^{-\Delta\Delta Ct}$ stated in the Methods. Please see example below:

MC1R $\Delta Ct = 12.43$

MC3R $\Delta Ct = 15.81$

$\Delta\Delta Ct = 12.43 - 15.81 = -3.38$

Then: $2^{-(-3.38)} = 10.41$

In the revised manuscript, we have added this formula in the Figure legend (1B and 3B) and provide the original reference article where the method is described in the Methods section (Ref 53).

Normalisation against the factor being measured is not usual practice – if normalisation is against HPRT, then another unchanging housekeeping genes should also be used: a large number of genes show altered expression on senescence so any hint of a senescent phenotype should require use for at least two normalisation standards in reverse transcription-qPCR.

Here we used *HPRT1* as a reference control gene. We are aware of the misuse of “reference genes” for real-time PCR data normalization (typically confused with the term “housekeeping gene”). Indeed, we published a methodology paper (PMID: 24493325) where we elaborate on this problem and address the inconsistencies it may cause. In the same study, we provide experimental data for the identification of the most appropriate reference control for PCR normalization in arthritic tissue sample extracts.

The mere addition of another reference control is not a good thing *per se* (e.g. if both reference genes are skewed in the same direction, let’s say towards up-regulation, this will aggravate the problem and lead to erroneous results). By contrast, a detailed study of the behaviour of reference genes is more appropriate. We performed an initial test to determine the validity of *HPRT1* using the method mentioned earlier, that take into account the variability and stability of the expression values within and between groups. As reported in the table below, *HPRT1* showed remarkably high consistency within and between groups, making it suitable for normalization in our conditions. While *GAPDH* showed good consistency as well, *HPRT1* performed better showing lower variability between groups (see delta Δ value), and lower SD particularly in the vehicle group (reflecting lower patient heterogeneity than *GAPDH*).

HPRT				
	VEHICLE	BMS		
Mean CT	26.85	25.78		$\Delta=1.07$
SD	0.129	0.189		
GAPDH				
	VEHICLE	BMS		
Mean CT	19.03	17.83		$\Delta=1.2$
SD	0.378	0.15		

We are confident that the reference gene that we used was optimal. We have included in the Methods section the reference for the article (Ref 54 in the revised manuscript) explaining the procedure that we followed for the selection of the appropriate reference control gene.

Quantification eg of p53 levels in Fig 2C does not seem consistent with the gel shown.

The graph was constructed using the quantification of six independent experiments performed in different patient cell lines, and the fold increase with respect to control was shown. The gel provided was a representative one out of the six, and it was chosen to reflect the average

response of the 6 samples (i.e. increase for BMS –slightly higher on average for the dose of 3 μ M-, and very minor increase for α MSH compared to control, C). In any case, to satisfy the Reviewer request, we have modified this figure (figure 2C) to present the actual quantification values (instead of fold change) which allow us to include the values for the control group. Maybe it is clearer this way. See below:

The use of forskolin is not explained in the legend/accompanying text and it is left to the reader to piece together the rationale – similarly, the use of the ionophore is not explained, and the numerical value given on the graph does not correlate with the concentration stated in the Methods section.

We have now fixed this omission. Forskolin and Ionomycin are the gold standard widely used positive controls for cAMP and Ca flux determinations, respectively. We appreciate that the readership of our manuscript will be wide and will involve readers with no knowledge on signalling measurements. We have then added a note to the Methods (page 30-31) and Figure legends 1E,F to indicate that these compounds were used as positive control for the appropriate assay.

The numerical value stated in the graph is not the concentration but the actual expression value so that the reader can compare the results from the MC compounds with the one obtained with the positive controls. To avoid misunderstanding, we have removed these numbers from the graph and included them in the Figure legends (1E,F) instead.

There are errors of this type throughout, as well as odd phrasing and grammar, omission of definite articles in parts, incorrect use of singular instead of plural in occasion (e.g. line 559 'receptor'), and use of 'on' instead of 'of' in several parts. Such lack of care in proof reading does not inspire confidence in the care taken to conduct rigorous experiments, particularly when labelling of figures is missing (e.g. scale bars), units potentially incorrect (e.g. IL-6 is measured in μ g/ml which is extremely high – senescent fibroblasts usually secrete in the order of ng/ml or pg/ml of IL-6 into the culture medium). Parts of figures are incorrectly cross-referenced (e.g. line 883 – I presume G is meant, not F?) Use of 'and' instead of 'or' is deeply misleading (e.g. line 745, but multiple such cases throughout). Use of dashes instead of parentheses is also confusing especially when used with numerical values e.g. dasatinib dose line 844. Text and figure labelling are not always consistent eg CtrlS in fig and ConS in text. All gels require labelling of size markers.

- Grammar has been revised and the identified typos corrected.
- Figures labels and legends have been revised and rewritten to make them more comprehensible.
- The measurements for IL-6 do not correspond to senescence fibroblasts, but to fibroblasts stimulated with 10 μ g/ml SAA \pm MC drugs, for 24h, as indicated in the Figure legend (and now

also indicated in the corresponding panels in Figure 1H). We have revised the data and units and ng/ml are the correct ones. Besides this, synovial fibroblasts from RA patients are quite unusual with respect to their immune status, and typically produce large amounts of cytokines, several orders of magnitude greater than other fibroblasts like skin fibroblasts.

- Line 883 reads: “*included in this study (mean±SD)*”. We believe this reviewer is referring to line 833 instead. We have corrected the labelling in this line: “*(H) Activated caspase-3 was assessed by fluorescent microscopy on human macrophages treated as in panel G*”.

- “And” has been replaced by “or” through the text where appropriate.

- Dashes and parentheses have been revised.

- Gel size markers have been added to the Western Blot gels where they were missing.

Measuring senescence-associated beta galactosidase (SABG) in cell types such as macrophages cannot be used for assessment of senescence – their high lysosomal content gives a high baseline of SABG staining (as the authors observe in Fig 1F but no not comment on). The high baseline rate of macrophage apoptosis also calls into question the validity of the method used - are the cells excessively stressed?

We have conducted a new experiment to provide measurement of senescence by p16 expression, in addition to the SA-βGal staining. As seen in the new graph provided (Figure 2F) there was no difference in the % of p16^{INK4} positive cells after BMS treatment.

Regarding apoptosis, we observed moderate basal levels in the control group to ~25% AnxAV/PI double positive events. In our opinion, this is an expected observation as experiments were conducted with freshly prepared blood-derived monocytes which are then differentiated into macrophages using a standard 7-day protocol under M-CSF stimulation (see Methods). In our view, it is not surprising that a degree of cell death takes place in these settings, with cells living on a dish even for this period of time. In addition, we note that “stress” in immune cells is normally associated with delayed apoptosis (rather than increase) that allow immune cells to live for longer and fight the particular stressor (e.g. infection...). In all cases, in this set of experiments cells were incubated in the same conditions across the board, that is with or without BMS.

Since there is currently no single specific biomarker for senescence, the authors have sensibly examined a number of parameters including a proxy for proliferation rate (cell number, though method of assessment is not given even in methods), SABG, and p16, as well as looking at morphology/size for SFs. They later rely on single markers to determine senescence (either SABG or p16), neither of which is sufficient alone to show senescence: SABG is simply a marker of lysosomal stress, whereas p16 is elevated on cell cycle arrest, either transient or permanent, and is not always present in senescent cells (it is lineage specific).

We agree with the Reviewer, that is, there is no universal marker of senescence hence we tested multiple markers in our experimental settings to ensure that the cellular process we were observing on synovial fibroblasts upon BMS treatment was, with no doubt, senescence. The ‘panel of senescence markers’ included: cell proliferation, SA-βGal, p16, p21, p53, lysosomal expansion, presence of bi-nucleated cells, down-regulation of cyclins, up-regulation of pro-survival signals, and more, as presented in Figure 2 and Supplementary Figure 2.

Respectfully, we disagree that the presence of p16 on senescent cells is “lineage” specific, but rather it is dependent on the particular mechanism mediating senescence in a given environment (type of cell + type of stimuli). We demonstrate here that the type of senescence induced by BMS in synovial fibroblasts is p16 dependent and we also demonstrate that this marker fully correlates with SA-βGal staining (Figure 2D).

For this reason, we believed unnecessary (and quite unrealistic) to measure every time all markers in all the experiments when cells and conditions and type of stimulus were the same. We continued our studies using either SA-βGal or p16 when more appropriate (e.g. the enzymatic activity of SA-βGal is not preserved on formalin fixed paraffin embedded [FFPE] samples, and hence p16 was more appropriate for the joint samples).

Altogether, we are confident to have identified, and studied, SF senescence in a variety and complementary experimental protocols.

Figure labelling is inconsistent, as are treatments e.g. various doses of BMS are employed even within a single figure, but without any justification.

The majority of experiments were conducted at 1µM, chosen according to our large experience on the pharmacology of melanocortin receptor ligands. Often, we have been restricted to one concentration due to the limited availability of human arthritis cells and their short life (i.e. ~8 passages). However, in a few occasions we tested additional concentrations when more material was available. We note that within the range of 1-10µM, there is no substantial difference in the pro-senescence effect of BMS. We do not see the necessity to test additional concentrations within the 1-10µM concentration range (e.g. 3 µM).

We have addressed this issue, which is important by no mean, and ensured that concentrations used in each experimental setting are consistently indicated in the Figures and inserted in the corresponding Legends.

There are occasional oddities for which the logic is not presented e.g. why are adipose and liver cells included at the end of Fig 5?

In our view, any new therapeutic treatment needs to be tested, not only for the expected therapeutic effect on the target organ/tissue, but also for effects that it may have on other organs/tissues. This is crucial to define the beneficial bio-actions and also potential unwanted effects. Although it was not the main scope of this study, as material was available from the arthritis model, we initiated a screening on the potential pro-senescence effects of BMS on other tissues such as liver and adipose, as these tissues are important in the context of ageing.

We have expanded the paragraph related to these results to explain how we performed tissue screening to identify potential off-target effects. Indeed, upon request of Reviewer#3, we have expanded this screening to 7 different tissues, which is now presented on a new Supplementary Figure 4.

Perhaps my greatest concern lies in interpretation of the RNAseq data. A major premise of the paper is based on BMS-dependent changes in gene expression, with extensive bioinformatics work up (with a number of nice-looking but not necessarily deeply informative figures - though I appreciate a number of -omics papers are similar). Much is been made in the paper of 'marked' changes, 'major' upregulation or dysregulation, 'significantly altered' (line 798) but in all cases, the genes in question show fold changes of >0.5 and <2 i.e. not normally considered significant. Biologically, such changes may be important but statistically they are not: if a large number of factors in a pathway all change in the same direction in response to treatment, then that may be significant – proper statistical analysis of the pathways needs to be conducted. Fold changes of the cholesterol pathway are not even given.

ALL genes included in the study were statistically significant (i.e. $p < 0.05$), as this was the filter set to produce the list of 1,952 differentially expressed genes. Any non-significant gene (even if the FC was relevant) was completely removed from the study.

In any case, we agree that we had not explained properly our approach in the Methods. We have now included details of the filtering process both in the Methods (page 34) and Figure legend 3A.

ALL fold changes (as well as full names) for all genes included in the manuscript were given in the Supplementary Table 3 (Excel file), organised to state to which figure each gene corresponded, including the cholesterol pathway genes. The list includes 184 genes and we think it is unrealistic and ineffectual to include all those numbers in the figures. The supplementary table is the correct place for this information.

We believe the Reviewer did not have access to that excel table during the review process. A cropped portion of that table is shown as an example:

Supplementary Table 3. Extended information on genes identified in RNAseq.

Symbol	Gene name	FC	Refseq ID	Locus
qPCR validation (Related to Fig3A,B)				
ADAMT52	ADAM metalloproteinase with thrombospondin type 1 motif 2	0.79	NM_014244	chr5 : 179110850 - 179345430
CD74	CD74 molecule, major histocompatibility complex, class II invariant chain	0.74	NM_004355	chr5 : 150401636 - 150412936
CDH11	cadherin 11, type 2, OB-cadherin (osteoblast)	0.84	NM_001797	chr16 : 64943752 - 65122137
CDKN2A	cyclin-dependent kinase inhibitor 2A	1.09	NM_000077	chr9 : 21967751 - 21975133
COL3A1	collagen, type III, alpha 1	0.65	NM_000090	chr2 : 188974372 - 189012746
COL5A1	collagen, type V, alpha 1	0.72	NM_000093	chr9 : 134641804 - 134844842
CTS5K	cathepsin K	1.99	NM_000396	chr1 : 150796207 - 150808441
DMKN	dermokine	1.79	NM_001035516	chr19 : 35497216 - 35501911
HMGCR	3-hydroxy-3-methylglutaryl-CoA reductase	1.53	NM_001130996	chr5 : 75337167 - 75362101
HMOX1	heme oxygenase 1	1.58	NM_002133	chr22 : 35381066 - 35394214
HRAS	Harvey rat sarcoma viral oncogene homolog	1.12	NM_001130442	chr11 : 532241 - 535567
HSD3B7	hydroxy-delta-5-steroid dehydrogenase, 3 beta- and steroid delta-isomerase 7	1.41	NM_001142777	chr16 : 30985197 - 30989152
INSIG1	insulin induced gene 1	1.98	NM_005542	chr7 : 155297775 - 155310235
LDLR	low density lipoprotein receptor	2.04	NM_001195799	chr19 : 11089361 - 11133829
MMP11	matrix metalloproteinase 11	1.29	NM_005940	chr22 : 23772818 - 23784316
POSTN	periostin, osteoblast specific factor	0.80	NM_006475	chr13 : 37562581 - 37598844
PTEN	phosphatase and tensin homolog	0.89	NM_000314	chr10 : 87863437 - 87971930
TNC	tenascin C	0.75	NM_002160	chr9 : 115019575 - 115118257
Senescence profile (Related to Figure 3C)				
BAG1	BCL2 associated athanogene 1	1.12	NM_001172415	chr9 : 33252470 - 33264761
BB3C	BCL2 binding component 3	1.32	NM_014417	chr19 : 47220821 - 47231194
BCL2	B-cell CLL/lymphoma 2	1.58	NM_000633	chr18 : 63123345 - 63319380
BCL2L1	BCL2-like 1	1.39	NM_001191	chr20 : 31664451 - 31723098
CCND2	cyclin D2	0.91	NM_001749	chr12 : 4273735 - 4305556

In other tests (eg t test), it is necessary to state whether one or 2 tailed tests were used.

Unless otherwise indicated, t-tests are normally presumed one-tailed. We have specified this in our statistical analysis section (page 35).

In many instances (measurement of levels of cytokines, cholesterol, bile acids etc etc), box and whisker plots or violin plots would be much more informative than bar graphs.

Box and whisker or violin plots are appropriate graphs for the representation of descriptive statistics, normally used to describe large populations and how the data-points are distributed (mean/median, quartiles, range, etc). However, the data regarding cytokines, cholesterol, etc, etc... represent the effect of a particular action (i.e. drug treatment) on a sample taken from a particular population (i.e. inferential statistics) and hence, graphs that clearly show the estimation of the error are required. Box and whiskers and violin plots will be misleading because they do not show error bars.

In any case, to make our experimental data more immediate to the readership, we have converted all bar graphs into dot plots to show clearly data distribution while at the same time displaying mean and standard error, as per Nature Communications policy.

Since MMPs and ADAMs both remodel the ECM, how do the authors account for opposite changes in expression on BMS treatment being consistent with senescence and tissue repair?

We did not identify changes in ADAM genes in our study. We believe this Reviewer is referring to a different gene family, ADAMTS.

The increase of MMPs accompanied by decrease in ADAMTS proteins is as consistent as the typical increase in MMPs accompanied by increase in TIMPs (metalloproteinase inhibitors). Metalloproteases are usually co-expressed with their inhibitors in order to achieve a “balanced” response preventing excessive tissue damage produced by MMPs. The increase of MMPs together with the decrease of ADAMTS is reflecting a balanced response. Furthermore, although

they are both metalloproteases, they are not redundant proteins and many differences exist between them. Importantly, ADAMTS are recognized as the major drivers of aggrecan degradation and hence cartilage destruction, while the role of MMPs in aggrecan degradation in RA is much less important. Hence, in the particular context of arthritic synovial environment, the decrease in ADAMTS enzymes could be extremely beneficial for tissue protection (i.e. the cartilage).

We have added a paragraph in the Discussion (page 19) to comment on this.

Where a patient cohort of 20 individuals is used, it is not helpful to state that 5% show a particular haplotype – that represents just one individual. Instead, the number of patients from the cohort should be given as with this sample size, % is misleading. On a further point of statistics, it appears odd that the symptom duration standard deviation is far greater than the mean – it is really the case that there is a +/-10 year SD in a cohort who have suffered symptoms for ~6 years? That implies some patients have been included who have suffered for a large negative time period?

Frequency data is always given as percentages as this allows a direct comparison between different works and the value *per se* has a universal meaning, i.e. 5% always suggests low frequency, 90% always indicates high frequency, independently of the population size. Providing the actual number will require the reader to do the maths for each study according to the population sizes to be able to compare different studies. In any case, we have added an extra column to the right (Figure 7B) to indicate the number of patients for each haplotype, together with the percentage.

Regarding the second comment, the key point to understand these numbers is that patients have suffered symptoms for ~6 years... **on average (mean)** and the large SD obtained can be envisaged by the large range (provided in Supplementary Table 1). Below are the individual numbers for all patients:

Symptoms duration (weeks): 10, 5, 4, 6, 9, 10, 4, 1040, 364, 1040, 1560, 1560, 156, 1040, 10, 4, 6, 1, 3, 7.
Mean: 342
SD:558.7

We have double-checked our calculations and we can confirm that they are correct. We also modified the legend to this Supplementary Table 1 to indicate that data refer to the time of sample collection, in the event this was not clear.

The Methods section is written in too brief a manner to allow other labs to repeat the studies e.g. antibody dilutions are given but not clone number or other unique identifier; % are given where molarities are required (eg glutamine in culture medium).

The antibody details including clone numbers were given in the “Antibodies” section of the Reporting Summary that will be attached to the article in its final version. In this version of the manuscript we have added further details throughout the entire Methods section.

Image processing states that exposure was optimised for printing – but it is critical that gain and exposure times are kept constant for immunofluorescence to allow direct comparison between samples and controls.

Images were taken all at the same gain and exposure, and this chosen gain and exposure were selected to provide images at the right quality for publication.

We have re-phrased that sentence as, clearly, it was misleading since we did not mean to convey the notion that each image was taken at different conditions (page 33).

While the Discussion mentions improvements in OA on senolytic therapy (ie the opposite of findings present here in RA), it could benefit from more balanced presentation of where senescence may be beneficial and where it is detrimental - eg idiopathic lung fibrosis is senescence-associated and senolytic treatment appears helpful. Acute wound healing needs acute senescence while chronic senescence blocks wound healing.

We elaborated deeply into the role of senescence in OA as it has important implications in the context of the present work on RA. The dual role of senescence and when it is good or when it is bad is a very long debate and would involve discussions in the fields of cancer, ageing, tissue fibrosis, repair and inflammation.

We cannot hope to cover this (exciting) discussion to the extent it would deserve within the space of the manuscript, and it may be the focus of a future review. However, since truly important as a discussion topic, we have extended the Discussion (page 22) on two major key points that determine the positive and negative (good and bad) outcome of cell senescence: **i)** in which cell type senescence is happening, i.e. parenchymal cells (functional tissue of the organ) or mesenchymal cells; and **ii)** the role of impaired immune surveillance, recently published in this journal, Nature Communications, as the responsible for the detrimental effects of senescence due to a prolonged effect of the SASP.

Overall, I think there may be real value in this work, and the authors may have come upon a very exciting potential new therapy for RA with an additional avenue for patient stratification and hence precision medicine in RA, but the manuscript and figures need major tidying up, tightening and improving in scientific precision before the paper is suitable for publication.

We thank this Reviewer for sharing this view while at the same time appreciating the value for patients that it may have. As requested, we have incorporated the changes suggested. We are confident these changes have greatly contributed to improve our manuscript and we are grateful to this Reviewer for the thorough revision of our work.

Reviewer#2

Montero-Melendez and co-workers studied the melanocortin (MC) in rheumatoid arthritis in humans and animals. They showed for the first time that MC type 1 receptor (MC1) are need to induce a cellular senescence phenotype characterized by arrested proliferation, metabolic re-programming and marked gene alteration resembling the remodeling phase of wound healing, with increased MMP expression and reduction in collagen production. This could be a novel way to control joint inflammation and arthritis.

This study is very interesting, experiments are well planned, methods and statistical analysis seem appropriate, results are clear and the references are updated.

I have the following minor comments:

1. The paper should be shorter

We have revised the manuscript and simplified parts whenever possible. However, to satisfy the requests by other Reviewers, we have had to add additional information and details but this was mainly done in the Methods section and Figure legends to avoid extending the length of the article, as we agree with this Reviewer on the importance to keep the manuscript focused and amenable to the reader.

2. Please explain better GPCR?

We have elaborated a bit more on GPCRs and their importance for drug discovery, and added references for further reading (Discussion, page 16).

3. Introduction: authors should be quote other important effects of melanocortins (e.g.: protective effect of MC on circulatory shock, myocardial and cerebral ischemia, experimental Alzheimer disease) and respective groups of research.

We have included an extra paragraph in the last part of the Introduction (page 5) to include this request. Clearly this has extended the word count.

Thank you for pointing this out.

Reviewer #3

In this manuscript, Montero-Melendez et al. reported a novel finding that activation of the melanocortin type 1 receptor (MC1) by a compound named BMS-470539 (BMS) can induce a senescence phenotype in synovial fibroblasts. This senescence phenotype includes proliferation arrest, lysosomal expansion, SA-bGal and p16INK4 staining and genome-wide gene expression changes. Furthermore, the authors tested the functional importance of this RF senescence induced by BMS in a K/BN arthritis model. The manuscript also discussed the effect of MC1R variants on BMS-induced RF senescence. Overall, the main finding (BMS induces a senescence phenotype in synovial fibroblasts) is novel and interesting. However, with the consideration of the below major points, the results are mostly descriptive and preliminary without much mechanisms revealed. Importantly, the data quality in the arthritis model is poor, and the results did not fully support the authors' interpretation. Further, some important outcomes from senescence, such as SASP (senescence induced secreted factors, cytokines) and inflammation, are not examined in this study. It is therefore not clear for the long-term effects of BMS-induced RF senescence on RA.

We thank Reviewer #3 for the time invested in revising our manuscript. To address their important criticisms and constructive suggestions, we have performed several new experiments, provided new high-quality images for the histological analyses and addressed all other comments as detailed below. We acknowledge the Reviewer's comments and suggestions and we believe they undoubtedly contribute to a general improvement of the manuscript and our work. Please find below our point-by-point answers:

Major concerns:

1. One critical feature associated with senescence is SASP (senescence induced secretome, for examples growth factors and cytokines). The secreted factors and cytokines often trigger immune response (both innate and adaptive) and chronic inflammation, which may deteriorate RA inflammation. However, the authors did not test this important feature, so it is hard to judge the therapeutic importance of long term BMS effects on RA.

This is an important point that we considered early on as the potential effects of senescence associated secretory phenotype induced by BMS are fundamental to understand the outcome (beneficial or not) of the whole process. In our analyses, we did not observe any BMS-mediated increase in the "pro-inflammatory status" of our fibroblasts as measured either by ELISA in supernatants or in the RNAseq profiling. There are two reasons that might explain this:

1) Synovial fibroblasts from RA patients greatly differ from other fibroblasts (e.g. skin) in their inflammatory profile. They secrete large amounts of cytokines and other factors *spontaneously* and these promote leukocyte recruitment, induce angiogenesis and cause cartilage degradation (PMID: 20193003, PMID: 20739221). This pro-inflammatory status derives from an epigenetically imprinted behaviour explaining why these cells remain highly activated *in vitro* even when cultured under resting conditions. We believe that senescence on these cells does not lead to any further activation, as we observed in our experiments, because they are already highly activated.

2) Although it is normally generalized that SASP is pro-inflammatory, it has been reported that the nature of the SASP is strongly stimuli/context dependent hence it is not always associated with a pro-inflammatory profile (e.g. PMID: 21880712, PMID: 26867806), a notion consistent with our observations. We have now included the data on cytokines release requested by this Reviewer (Supplementary Figure 1K), demonstrating that the SASP produced in our experimental setting was not pro-inflammatory, as suggested by others' research too. We indeed observed a modest decrease on the levels of some of these mediators, likely reflecting the effect of BMS on proliferation.

Results (new Supplementary Figure 1K) and Discussion (page 17) have been extended accordingly.

2. The off-targeting effects of BMS in vivo: although the authors did not find BMS induced senescence in liver and adipose in the arthritis model, did they examine other tissues or organs, such as kidney, spleen, brain, lung and skin? Because they found BMS can induce senescence in human dermal fibroblasts, the senescence effect of BMS seems broad not limited to synovial fibroblasts.

Yes, our data indicate that the application of the pro-senescence effect of BMS might extend beyond the synovial environment. We have observed similar pro-senescence effect in skin fibroblasts *in vitro*.

This finding could be of importance as it may imply a completely novel therapeutic area for intervention to harness the presented pro-senescence approach for the management of fibrotic diseases, in which over-activated fibroblasts are responsible for tissue dysfunction and eventual failure. The profile we observed with BMS in the present study strongly suggests an anti-fibrogenic potential due to the pro-repair profile (reduction in collagens, increase in remodelling) that BMS-induced senescent cells acquire.

On the suggestion of this Reviewer, we have performed another set of experiments to expand our screening to include brain, fat, kidney, liver, lungs, skin and spleen. To this end, mice were treated for 6 days with BMS as in the arthritis experiment. Using this dose and treatment schedule, we have not detected any increase in p16 cells in any tissue apart from the articular joint.

All these new data are included in a new figure, Supplementary Figure 4.

3. Although the group performed profound bioinformatic analysis, there is little data to experimentally elucidate the mechanisms suggested by these bioinformatic information.

We conducted the RNAseq analysis as an actual validation of many of our experimental observations (i.e. our major findings reported herein were not acquired *via* gene analysis but via actual experimentation). The RNAseq analysis confirmed that the overall gene expression profile of BMS-treated SF was entirely consistent with the observed induction of senescence, lysosomal expansion, reduction in proliferation, and reduced pro-inflammatory status. Then the RNAseq revealed something new, the strong up-regulation of cholesterol pathway, a dataset that prompted us to conduct multiple experiments to validate this observation (all Figure 5).

Prompted by the Reviewer, we have dwelled further into the RNAseq analysis and have identified a new dataset and mechanism that was not presented in version 1 of the manuscript: the inhibition of the Notch pathway. We have experimentally validated the involvement of this pathway in the pro-senescence effect of BMS; we believe these new data add an important mechanistic insight in the actions downstream selective MC₁ activation, because as discussed in the manuscript, inhibition of Notch is known to be beneficial in RA. We now link this mechanism to SF senescence and the anti-arthritic properties of BMS and, potentially, novel MC₁ selective agonists.

In summary, we identified a consistent down-regulation of the Notch pathway from the RNAseq dataset, and we demonstrated that the expression of the receptor Notch3, at the protein level (immunofluorescence), was down-regulated by BMS. According to this, we hypothesized that the activation of Notch may then prevent the pro-senescence action of BMS, a hypothesis that we demonstrated to be true by testing the effect of a recombinant Notch ligand (DLL4), which abrogated the induction of senescence by BMS. All these new data are presented in the new Figure 4, and discussed in page 19.

4. The bile acid secreted by SF induced macrophage apoptosis. With consideration of the importance of macrophages in RA, does this macrophage apoptosis affect inflammation and immune states of the arthritis mice?

As the Reviewer rightly comment, macrophages are also pathogenic cells in RA and co-exist with fibroblasts in the synovial lining. It might then be possible that indirectly, the SASP released by senescent SF will influence neighbouring macrophages. However, to fully understand this, extensive work will be necessary because macrophages highly express MC₁ receptor and hence, direct actions from BMS will lead to additional effects on these cells (e.g. modulation of cytokine release, see for example PMID: 30154720) regardless of direct and/or indirect effects through the SF. This will be an interesting area to explore in a separate study as it lies out of the scope of the present manuscript (already very extensive). In any case, we have considered this point in the Discussion section (page 20-21).

5. Since BMS activates MC1, have the authors tested the general effect of MC1 activation in brain function--the food intake regulation, in vivo in the arthritis model?

The melanocortin receptor that regulates food intake is MC₄, and to a lesser extent MC₃, both expressed in the hypothalamus and other brain regions. To the best of our knowledge, it has not been reported any role for MC₁ on food intake regulation. In our experimental settings (with a relatively short duration treatment of 6 days) we did not observe any effect on body weight on mice treated with BMS compared to control mice.

We can also share with the Reviewer that our initial analysis on the 'druggability' of the BMS molecular structure indicates that this compound should not cross the blood-brain barrier and hence it might be devoid of any central effects.

6. The evidence of lysosomal compartment is not clear.

We provide new images in revised Supplementary Figure 1A showing presence of a large lysosomal compartment in a clearer fashion. We hope.

7. Have the authors tested the apoptosis of SF in response to BMS?

We did not measure apoptosis in our settings although we performed the Alamar blue cell viability assay, showing a substantial increase in metabolic activity on cells treated with BMS compared with untreated SF, suggesting that cells were not dying at all, a conclusion that is consistent with a pro-senescence state. This result is included in Supplementary Figure 1B. In addition, in our RNAseq analysis we detected up-regulation of several genes that transcribe anti-apoptotic proteins such as *BAG1*, *BCL2* and *BCL2L1*: this is shown in Figure 3C and Supplementary Figure 2A.

To add further evidence, we have now analysed expression of active caspase-3 and no difference was detected in BMS-treated cells compared to control cells. These new data have been included in revised Supplementary Figure 1C.

8. Many experiments lack appropriate controls. For example, Fig. 1C should have controls, including skin fibroblasts and OA fibroblasts. Fig 1E, no positive controls provided.

Fig 1C: We think we have labelled the graph in a confusing way. Apology. This graph presents a set of descriptive data on the levels of ACTH released by SF. But the way we labelled it seemed we had *treated* the cells with ACTH.

These data then represent a description of the production of this mediator from SF, and we believe it is not relevant to the present study to quantify what could be the levels of ACTH produced from other types of fibroblasts. We have corrected the figure to remove this confusion: in revised Figure 1C, the dotted line represents the negative control (i.e. ACTH quantified in media alone without cell incubation)

Figure 1D: Forskolin is the positive control for this assay. We have added a note to the Methods (page 30) and Figure legend 1D to indicate this.

9. Some data quality is poor: the western blot bands in Fig 1D are blurry.

We have noticed that the quality of some images in the low-resolution file provided for Reviewers is lower than our original figures. Below are the blots with the bands for Fig1D we submitted, compared to the one that reviewers received. We will ensure during the proof-correction that the quality is preserved.

The quality of histological slices in Fig5 F and G is poor; it is hard to tell the details of the structures and generally the 4x pictures should be provided too. The locations in F seem different between conditions, and tissues lost integrity. The criteria of synovitis, such as immune cell infiltration, pannus formation extent and tissue destruction (cartilage and bone), was not clear although the scores 0 (no synovitis), 1 (mild) and 2 (moderate) were mentioned. The BMS group in G shows more bone destruction, which is against the interpretation of the data. Some pictures show more joints, while the others only show surface of the joints. The tissue locations shown in different conditions are not comparable.

This is a good point, thank you. We have now repeated the histological analyses and acquired higher resolution images. As before, images were taken at the same location, but we have now re-oriented them to make clearer for a non-trained eye that they correspond to the same part of the tissue. We also provide whole-tissue images. These new images are shown in Figure 6G, as well as Supplementary Figure 3.

The term synovitis is a quite generic one. It is typically used to refer to the extent of leukocyte infiltration into the surrounding tissue, which can be identified by H&E staining. Using this term, we do not include “pannus formation” or “tissue damage”, as these are addressed separately. In this acute model of inflammatory arthritis (9 days), bone damage is not usually seen as in other long-term models like CIA, and hence we do not score for them. To observe bone damage in the K/BxN model of inflammatory arthritis one has to run active arthritis for over 20 days (PMID: 15334485). However, we do detect cartilage damage measured by loss of proteoglycans (toluidine blue staining), but we score this marker separately from “synovitis”. To avoid misinterpretation, in v2 of the manuscript we have now removed the term “synovitis” and use for this set of analyses the term “cell infiltration”.

What this Reviewer indicated as bone destruction in the BMS group in old Figure G, corresponds to the bone marrow zone and artefacts of sectioning rather than *bona fide* bone destruction; as mentioned earlier this acute model is not associated with bone destruction. To avoid misunderstanding, we have removed the safranin-O staining and present only the toluidine blue on the knee sections to address cartilage integrity.

10. Some experiments or results lack critical information to understand: How was BMS administrated in vivo (orally, ip, iv)? What are those “other members of the MC pathway” in Fig 1A? In Fig 4D, what are the specific EIA? Suppl Fig 1, what are the details of the cell culture?

BMS was administered ip, as indicated in the Methods section. We have now added this information also to the Figure 6 legend to facilitate interpretation.

We have expanded a paragraph in the first section of Results (page 6) to indicate what are the melanocortin genes.

All the ELISA and EIA kits used in this study are explained in the Methods section (page 30) under the heading “**Enzyme-Linked Immunosorbent Assay (ELISA)** and **Enzyme immunoassays (EIA)**”.

The cells used in Supplementary Figure 1 are human fibroblasts, melanocytes and HEK cells. The culture conditions of these cells are the same as used in Results presented in main Figures and were included in the corresponding sub-sections of the Methods.

11. The information about the synovial fibroblasts used in this manuscript is confusing. Are these all from the 20 RA patients? or additional patients included? Some experiments used n<20 (for example, in Fig 1A, B, C. In Fig 3, n=8 donors, what are these 8 donors?), so how did the authors select the SF cells for different experiments?

We apologize for this confusion. Where it reads “donor” should say “patient”. We did not use additional patients over the 20 that we report in Supplementary Table 1. This mistake has been corrected.

These short-lived primary cells last for only 8 passages in culture (as indicated in Methods) and there is not enough material to conduct all the experiments with all patients. Unlike skin fibroblasts, synovial fibroblasts are a very precious and scarce material. However, the most relevant experiments supporting the main discoveries of the work presented here (i.e. induction of senescence and reduction of proliferation, as well as *MC1R* genotyping and cholesterol measurements) were conducted in the full set of 20 patients.

12. The introduction did not provide necessary background of previous studies on melanocortin system, senescence, and their relation with health and diseases, in particular with inflammation and arthritis (both RA and OA), especially the recent article published in Nature medicine regarding local clearance of senescent cells that attenuates OA, which provides proof-of-concept evidence of the importance of clearing senescent cells in OA.

We have now considerably expanded the Introduction section to introduce to the reader the concept of senescence and its well-known dual role in health and disease (i.e. beneficial or detrimental) in page 3. Similarly, a more extensive background has been provided describing the potential of melanocortins as a new therapy and more specifically for arthritis (page 5), in line with the focus of this work.

The Nature Medicine article mentioned by this reviewer on the clearance of senescent cells in OA was substantially covered in our Discussion where we highlighted that the protective effect is mediated by the elimination of senescence cells. We think that this study is sufficiently, and properly, considered in the Discussion (see below; page 22) and there is no need to introduce it early on. In our view, it helps focusing the Discussion on joint diseases.

“Interestingly, a senolytic rather than a pro-senescence approach has been tested in experimental osteoarthritis (OA)⁴⁴. The authors used the elegant trimodality-reporter mouse model p16-3MR, engineered to direct expression of pro-apoptotic proteins under the promoter of p16INK4 gene, thus inducing apoptosis selectively on senescent cells⁴, to reveal protection on cartilage integrity by the elimination of senescent cells. As known, there are fundamental differences between RA and OA including their relation with ageing. For OA, age is one of the most important risk factors. Senescence in chondrocytes may play an important role in OA pathogenesis...”

Minor points:

1. Details of RNAseq analysis are missing. It is insufficient to only list the names of bioinformatics resources.

The detailed analysis of the RNAseq raw data analysis was detailed in the Methods section “**Gene expression analysis**”. A latter section titled “**Bioinformatic resources and databases**”, listed again the tools used in the previous raw data analysis and then we added additional tools used for the functional analysis, to put them all together. As obviously this has resulted in some confusion, in v2 of the manuscript, we have left the raw data analysis in the corresponding section and renamed the second section as “**Functional analysis and databases**” (page 34-35). Further details on the tools used have been added. Of note, several of the resources listed are merely databases and not analysis tools, and so there are not really further details to include, apart from the version.

2. It should be helpful to understand the dataset if the authors could describe results more clearly and provide more details.

We have added considerable additional information on the Methods section as well as further details in the Figure legends to help in the interpretation of the results. In addition, we have revised the main text to clarify several other points. Thank you for prompting us to do so.

Reviewers' Comments:

Reviewer #2:

Remarks to the Author:

Montero-Melendez and co-workers have answered in appropriate manner to my comments. The paper is great improved and I have not any other comments.

Sincerely

Daniela Giuliani

Reviewer #3:

Remarks to the Author:

The authors' explanation to Q9 on the bone destruction is not appropriate. The benefit of KBN model is that this model can bypass the initial innate immunity phase to allow to study functional phase including tissue destruction in a faster way than other models, such as CIA. Many literature show bone destruction occur in tarsal joints around 10 days. The KBN serum strength and application details might affect the time course, but the authors can not conclude that the bone destruction can not be observed early around 10 days in this model. A bone biologist or pathologist should be included to evaluate the slices for bone destruction.

The authors successfully addressed other questions.

Reviewer #4:

Remarks to the Author:

In this manuscript, Montero-Melendez et al. provide evidence that selective activation of the GPCR MC1 promotes synovial fibroblast cellular senescence and a unique SASP profile that appears to have beneficial effects in the RA synovium. The approaches used to induce cellular senescence specifically in synovial fibroblasts may lead to novel approaches to control joint inflammation and RA. Since the previous review, many of the issues raised by reviewers have been adequately addressed and the manuscript has been significantly improved based on the actions and changes in response to previous reviewer comments. However, there are still additional issues that need to be addressed.

1. In regards to the interpretation of the RNAseq data, the authors state that ALL genes included in the study were statistically significant (i.e., $p < 0.05$), as this was the filter set to produce the list of 1,952 differentially expressed genes. However, there remain multiple concerns regarding the approach. For example, the authors do not comment on the criteria used to establish whether or not a particular gene is "expressed" based on RPKM or counts. For example, some have suggested using a RPKM threshold of 0.3 whereas others have used an absolute median gene count threshold of 10. Rationale should be provided for the expression threshold used. Furthermore, a false discovery rate (FDR) needs to be applied. This will reduce the number of genes that to those that were actually differentially expressed. The fold changes reported for some of the senescence genes in Fig. 3C are very small. For example, CDKN2A (p16) is only upregulated 1.09 fold, which is hardly changed if even at all.

2. The reliance on antibodies to mouse p16Ink4a is very problematic in the senescence field as all of the antibodies that are currently available from commercial vendors lack specificity. For example, the Santa Cruz antibody used throughout the literature (sc-1661) has been shown to detect a persistent signal by IHC even after p16Ink4a knockdown (PLOS One 8:e53313, 2013). Furthermore, the Abcam antibody used in this paper has also not been well validated, as the company does not provide negative controls in their validation. Further, the authors do not provide data regarding specificity. Because essentially all of the conclusions drawn from the mouse study

regarding cellular senescence (Fig. 6; Supplementary Fig. 4) are dependent solely on the use of the Abcam mouse p16Ink4a antibody, the authors need to provide additional in vivo evidence that BMS is, in fact, inducing cellular senescence specifically in synovial fibroblasts (Fig. 6B), and that treatment with senolytics (dasatinib plus quercetin) eliminates senescent synovial fibroblasts in vivo (Fig. 6D). Combinations of more specific senescence biomarkers in the same samples are necessary. Quantification of the data and proper statistics should also be provided.

3. Details regarding the dasatinib plus quercetin treatment regimen are lacking. For example, the doses utilized are quite different than those used in REF 27. Has the dosing regimen (days 4, 6, 8) been used in any previous studies? According to the Figure Legend (Fig. 6C), these drugs were administered intraperitoneally. However, in REF 27, as well as in all other previous studies administering these drugs as senolytics in vivo, they are delivered to animals by oral gavage. This needs to be explained. How do the authors exclude potential non-senolytic off-target effects of these drugs, especially given the short duration of the study and the short interval between dosing days?

4. Another concern is that the authors have not examined the long-term effects of activating senescent synovial fibroblasts in any in vivo models. While beneficial effects might be observed acutely, this approach may prove to be detrimental over time. This needs to be tested before any therapeutic potential of this approach can be further considered.

Rebuttal Letter #2

Reviewer#2

Montero-Melendez and co-workers have answered in appropriate manner to my comments.

The paper is great improved and I have not any other comments.

Sincerely

Daniela Giuliani

Reviewer#3

The authors' explanation to Q9 on the bone destruction is not appropriate. The benefit of KBN model is that this model can bypass the initial innate immunity phase to allow to study functional phase including tissue destruction in a faster way than other models, such as CIA. Many literature show bone destruction occur in tarsal joints around 10 days. The KBN serum strength and application details might affect the time course, but the authors can not conclude that the bone destruction can not be observed early around 10 days in this model. A bone biologist or pathologist should be included to evaluate the slices for bone destruction.

The authors successfully addressed other questions.

The joint sections have now been assessed by an expert in osteoarthritis and joint disease. The analysis of bone destruction has been conducted following the detailed scoring method described by *Pettit et al, American Journal of Pathology 2001,159:1689-1699*, in which the same mouse model (K/BxN serum transfer model) was used.

As seen in the graph below, although some bone destruction can be appreciated, only one or two mice per group displayed signs of bone damage, including the vehicle un-treated group, and the damage observed was generally very mild. This indicates that bone destruction cannot be used as a reliable measurement to evaluate drug efficacy in this acute model of arthritis (day 8) as, in our experimental settings, most of the mice do not show signs of bone damage even in the control group of arthritic mice.

We have added a line to the Methods section (page 29) to indicate that bone damage observed was minimal and inconsistent to be used as a marker of drug efficacy.

(Of note, this model does not bypass the innate immunity phase but rather the adaptive immunity phase typical of the first 2 weeks of the CIA model).

Reviewer#4

In this manuscript, Montero-Melendez et al. provide evidence that selective activation of the GPCR MC1 promotes synovial fibroblast cellular senescence and a unique SASP profile that appears to have beneficial effects in the RA synovium. The approaches used to induce cellular senescence specifically in synovial fibroblasts may lead to novel approaches to control joint inflammation and RA. Since the previous review, many of the issues raised by reviewers have been adequately addressed and the manuscript has been significantly improved based on the actions and changes in response to previous reviewer comments. However, there are still additional issues that need to be addressed.

1. In regards to the interpretation of the RNAseq data, the authors state that ALL genes included in the study were statistically significant (i.e., $p < 0.05$), as this was the filter set to produce the list of 1,952 differentially expressed genes. However, there remain multiple concerns regarding the approach. For example, the authors do not comment on the criteria used to establish whether or not a particular gene is “expressed” based on RPKM or counts. For example, some have suggested using a RPKM threshold of 0.3 whereas others have used an absolute median gene count threshold of 10. Rationale should be provided for the expression threshold used. Furthermore, a false discovery rate (FDR) needs to be applied. This will reduce the number of genes that to those that were actually differentially expressed. The fold changes reported for some of the senescence genes in Fig. 3C are very small. For example, CDKN2A (p16) is only upregulated 1.09 fold, which is hardly changed if even at all.

The criteria for filtering *expressed from non-expressed genes* was already explained in the Methods section under the heading ‘Gene expression analyses’: “*The read count data was filtered to keep genes that achieve at least one read count per million (cpm) in at least four samples*”. In other words, for any given gene to be considered expressed, it needs to reach at least $cpm=1$ in a minimum of 4 samples.

We have elaborated more on this sentence to make it clearer. It now reads (page 34):

“The read count data was filtered to keep genes that achieve at least one read count per million (cpm) in at least four samples. Hence, to be considered expressed, any given gene needs to reach a value of at least $cpm=1$ in at least the 25% of samples (i.e. 4 samples)”.

As this fourth Reviewer pointed out, we observed a fold change for CDKN2A of 1.09, which was confirmed by quantitative PCR obtaining a similar value of 1.13 (Figure 3B).

We wish to elaborate on this dataset and its implications.

1) As shown on the right graph on Figure 2B, BMS induced senescence on ~60% of the cells treated at 10 μ M for 7 days (the same protocol applied for cells used in the RNAseq experiment). This implicates that changes in gene expression are underestimated, as they will get *diluted* with the non-senescent cells which represent ~40% of the total. Importantly, immunofluorescence staining for p16 confirms the up-regulation at the protein level in senescent cells (Figure 2D).

2) Due to this inevitable biological effect, we decided to apply a less stringent approach for the RNAseq analysis to maximize the information generated from the sequencing, by preventing false negatives but accepting a higher degree of potential false positives. In fact, because our functional analysis was based on detection of patterns, networks, clusters and our interest centred on biological processes and pathways, we think this approach was appropriate. (We agree that, for instance, this may not have been appropriate for a gene-centric approach aiming to identify biomarkers, in which case it would be more advisable to avoid false positives, at the expense of missing out some genes).

3) On these premises, and accepting the potential presence of false positives, we performed a stringent experimental validation. It was pleasing to find that the major patterns and pathways that emerged from the RNAseq and that included i) metabolic alterations in the cholesterol pathway, ii) down-regulation of the Notch pathway and iii) and ultimately potential anti-arthritic actions were all experimentally validated. These sets of data are presented in Figures 4, 5 and 6, demonstrating that our “relaxed” approach (i.e. $p < 0.05$) was useful and valid for the detection of relevant biological functions. Hence, we believe the main conclusions of our study remain as they are underpinned by functionally validated data and do not derive from the mere identification of specific genes in the RNAseq experiments.

In line with this Reviewer, we agree definitively that there isn't a one-size-fits-all approach for RNAseq analyses. The analysis approach chosen for different experiments should be tailored according to the particular objectives of the study, nature of the data, expected heterogeneity levels and so forth. We believe the approach we undertook was the most appropriate for our purposes after evaluation of the particular conditions of our experiments and initial evaluation of the data.

The full data set is publicly available (GEO: GSE98658) for researchers who may wish to re-analyze our data using other conditions more appropriate for other goals, for example the identification of biomarkers of senescence (which was not the scope of our present work).

2. The reliance on antibodies to mouse p16Ink4a is very problematic in the senescence field as all of the antibodies that are currently available from commercial vendors lack specificity. For example, the Santa Cruz antibody used throughout the literature (sc-1661) has been shown to detect a persistent signal by IHC even after p16Ink4a knockdown (PLOS One 8:e53313, 2013). Furthermore, the Abcam antibody used in this paper has also not been well validated, as the company does not provide negative controls in their validation. Further, the authors do not provide data regarding specificity. Because essentially all of the conclusions drawn from the mouse study regarding cellular senescence (Fig. 6; Supplementary Fig. 4) are dependent solely on the use of the Abcam mouse p16Ink4a antibody, the authors need to provide additional in vivo evidence that BMS is, in fact, inducing cellular senescence specifically in synovial fibroblasts (Fig. 6B), and that treatment with senolytics (dasatinib plus quercetin) eliminates senescent synovial fibroblasts in vivo (Fig. 6D). Combinations of more specific senescence biomarkers in the same samples are necessary. Quantification of the data and proper statistics should also be provided.

The issues with the Santa Cruz antibody commented by this Reviewer are irrelevant to us because we have never used that antibody (indeed, we never use Santa Cruz's). The Abcam antibody we used, however has been recently validated in this same journal, Nature Communications, where Ovadya *et al* showed convincing co-staining of this same antibody with the other major senescence marker SA- β Gal in

mouse liver tissues (*Nat Commun* 2018 Dec 21;9(1):5435. PMID: 30575733). This validation is very reassuring and this article was already referenced in the link for the Abcam mouse p16 antibody provided in the Reporting Summary.

To provide further evidence of the validity of this antibody, we conducted additional experiments using a non-commercially available antibody, kindly provided by the senescence expert Dr. Manuel Serrano, which has been validated in KO mouse tissues (validation file can be accessed at <https://www.cnio.es/en/research-innovation/services/monoclonal-antibodies/>). We then tested the Abcam antibody against this KO-validated one to compare patterns of staining, using joint tissues from BMS-treated mice. As seen in the figure below, both antibodies produce the same pattern of staining in the synovial lining, and the merge image clearly shows a remarkable co-staining, strongly indicating that our antibody is specific for p16. Indeed, the KO-validated one (CNIO) donated by Dr. Serrano produced a bit more of background than the Abcam one.

Furthermore, we also tested the co-staining of both antibodies on UV-induced senescent cells using the mouse cells lines 4T1 (known to express p16, PMID: 28430642) and NIH 3T3 (known to be deficient on p16, PMID: 7585567). As it can be observed (figure below), p16 was detected by both antibodies with similar pattern of staining *only* on 4T1 cells, while only negligible staining was obtained in the p16-deficient NIH 3T3 cells.

4T1 (*p16* expressing cells)

NIH-3T3 (*p16* KO cells)

We are confident that we have used a validated antibody that specifically detects p16^{INK4}, and that our data are substantiated by this as well as complementary analyses: selective activation of MC₁ in fibroblast-like synoviocytes promotes senescence.

We have prepared a new figure (new Supplementary Figure 4) to include these results.

3. Details regarding the dasatinib plus quercetin treatment regimen are lacking. For example, the doses utilized are quite different than those used in REF 27. Has the dosing regimen (days 4, 6, 8) been used in any previous studies? According to the Figure Legend (Fig. 6C), these drugs were administered intraperitoneally. However, in REF 27, as well as in all other previous studies administering these drugs as senolytics in vivo, they are delivered to animals by oral gavage. This needs to be explained. How do the authors exclude potential non-senolytic off-target effects of these drugs, especially given the short duration of the study and the short interval between dosing days?

In our view, as Pharmacologists, specific characteristics of each model, drug and aims of the study need to be taken into account in order to design an appropriate efficacy testing. Simply because previous Authors have used a particular dose/regimen, these cannot be directly extrapolated to all other disease models. Zhu *et al* (Ref 27) used dasatinib+quercetin at 5+50 mg/kg, respectively. This was the case because they administered the drug once or twice monthly, and hence a high dose administration was appropriate. In addition, this approach was appropriate as they were evaluating a model in which senescence was already occurring as they used aged mice.

However, our disease model and study aims are very different: first, we needed to induce senescence with BMS by administering it daily from day 3 to day 8; second, to address the effects of senolytics, drugs ought to be administered regularly to match the pro-senescence action of BMS. Basically, one single shot would not answer the question we wanted to address, as the repeated administration of BMS (to achieve anti-arthritic effects) will induce further senescence, and consequently we might have risked to see no effect of the senolytic treatment on the pro-senescence actions of BMS. Therefore, we chose to use a lower dose and apply a repeated protocol of administration. Also, the low dose was selected because we used i.p. administration (hence greater bioavailability as the first-pass metabolism in the liver is avoided after enteric absorption). The intermittent protocol of administration was used to reduce the chances of potential off-target effects.

It is known that oral administration is a stressful administration route for drugs. Unless we are specifically interested in determining *oral efficacy* of a drug (which was not the case here) we avoid oral administration: this is i) to comply with the legal requirements on our Project License under the Animal (Scientific Procedures) Act 1986, ii) to avoid unnecessary additional stress to the animals and iii) to prevent risk of potential complications and further injury.

To exclude potential non-senolytic off-target effects of these drugs, we included a group of mice that were *treated only with senolytics* (light blue line in Figure 6C). We believe that the short duration and intermittent administration apply to this experimental setting contribute to the absence of side effects.

A new paragraph has been added to the Methods section to give more details regarding the schedule chosen (page 29-30).

4. Another concern is that the authors have not examined the long-term effects of activating senescent synovial fibroblasts in any in vivo models. While beneficial effects might be observed acutely, this approach may prove to be detrimental over time. This needs to be tested before any therapeutic potential of this approach can be further considered.

Evaluation of long-term effects of the BMS treatment *in vivo* will require long-term analyses that will represent a separate study on its own. It will require evaluation of different doses, different treatment schedules, evaluation of the co-administration of senolytics (as we comment in our discussion), evaluation in different models of arthritis, study of right time to give the drug (i.e. during early arthritis vs. established arthritis) and more. A totally different kettle of fish for a separate aim and story.

In our view, such a long term and costly study is beyond the scope of this manuscript and, if only, should also be performed on an improved molecule. A wise drug development programme would involve generation of a lead candidate molecule first, using the initial hit (BMS compound) as a starting point. Our druggability analysis on BMS pointed out several violations of Lipinski and Ghose rules including molecular weight and molar refractivity, as well as a not very favorable solubility. The most reasonable approach would be to first improve the structure of BMS through a medicinal chemistry program before conducting large scale efficacy testing and evaluation of long-term effects, which is costly not only economically but in number animals used for research, and hence should be fully justified and performed with a plausible candidate drug.

Definitely, this was not the scope of the present work, but it definitely should be a follow-up study.

We have added the following sentence (page 23) to highlight the need (in the future) of performing such a study:

“It remains to be seen what could be the consequences, and possible side effects, associated with long term treatment with this approach prior to further therapeutic developments”.

Reviewers' Comments:

Reviewer #3:

Remarks to the Author:

The authors put effort to carefully examine the bone damage in their mice. They did not observe consistent bone erosion, which is very likely because of the KBN serum strength, doses, and application times they used (the reviewer had a typo for "bypass the need to induce autoimmunity (not innate immunity) in this model"). Please add "in our experimental settings" after the wording "...the time course as a marker of drug efficacy." on p29. The reviewer has no other questions.

Reviewer #4:

Remarks to the Author:

The authors have adequately addressed my previous comments - thank you.

3. REVIEWERS' COMMENTS:

Reviewer #3 (Remarks to the Author):

The authors put effort to carefully examine the bone damage in their mice. They did not observe consistent bone erosion, which is very likely because of the KBN serum strength, doses, and application times they used (the reviewer had a typo for "bypass the need to induce autoimmunity (not innate immunity) in this model"). Please add "in our experimental settings" after the wording "...the time course as a marker of drug efficacy." on p29. The reviewer has no other questions.

As requested, the statement of "in our experimental settings" has been added.

Reviewer #4 (Remarks to the Author):

The authors have adequately addressed my previous comments - thank you.

We are pleased, thanks.